# TRIM25 targets p300 for degradation

Seham Elabd[1,2,*], Eleonora Pauletto[1,*], Valeria Solozobova[1], Nils Eickhoff[3], Nuno Padrao[3], Wilbert Zwart[3,†] ⓘ,
Christine Blattner[1,†] ⓘ

**p300 is an important transcriptional co-factor. By stimulating the transfer of acetyl residues onto histones and several key transcription factors, p300 enhances transcriptional initiation and impacts cellular processes including cell proliferation and cell division. Despite its importance for cellular homeostasis, its regulation is poorly understood. We show that TRIM25, a member of the TRIM protein family, targets p300 for proteasomal degradation. However, despite TRIM25's RING domain and E3 activity, degradation of p300 by TRIM25 is independent of TRIM25-mediated p300 ubiquitination. Instead, TRIM25 promotes the interaction of p300 with dynein, which ensures a microtubule-dependent transport of p300 to cellular proteasomes. Through mediating p300 degradation, TRIM25 affects p300-dependent gene expression.**

## Introduction

Protein levels are regulated at several nodes. One mode of protein level regulation acts through enhancing or reducing gene transcription. Gene transcription can be stimulated by binding transcription factors to promoters or enhancers of target genes and by posttranslational modifications of transcription factors and histones. Such modifications can be implemented by transcriptional co-factors such as p300.

p300 is, together with its paralogue CBP (CREB binding protein), among the most well-known co-activators for gene transcription. This co-activator function of p300 is, at least in part, mediated by its catalytic acetyltransferase domain that enables the transfer of acetyl groups onto lysine residues of target proteins and by the presence of a bromodomain that interacts with partner proteins (Chan & La Thangue, 2001). Target proteins of p300 are histone proteins, but also a vast number of transcription factors. Because of its co-activator function, p300 is involved in numerous cellular processes including cell growth and cell death (Chan & La Thangue, 2001). Perturbations in p300 activity are therefore also investigated for therapeutic interventions in the treatment of cancer (Welti et al, 2021; Chen et al, 2022). Despite its importance for gene transcription, little is known about the regulation of p300 abundance and activity.

TRIM25 is a member of the TRIM protein superfamily. This family is characterized by a specific domain structure: an N-terminal RING (Really Interesting New Gene) domain, one or two B-boxes and a coiled-coil region (Elabd et al, 2016). Through its RING domain, TRIM proteins frequently function as E3s and tag target proteins for degradation by cellular proteasomes. Some members of the TRIM family have also been shown to function as receptors in the autophagy pathway (Mandell et al, 2014). *TRIM25*, also known as *Efp* (estrogen-inducible finger protein, Inoue et al, 1993) is under direct transcriptional control of the estrogen receptor and is particularly abundant in estrogen-responsive female organs including the mammary gland. But at a lower level, the protein is also found in other tissues. TRIM25 mediates the transfer of ubiquitin and of ISG15 (interferon-stimulated gene 15) onto target proteins such as 1433σ or RIG-I (retinoic acid-inducible gene 1) and modulates protein half-life (Urano et al, 2002; Zou & Zhang, 2006). TRIM25 is involved in growth control and metastasis of cancer cells and in the defense against viruses (Martin-Vicente et al, 2017; Walsh et al, 2017; Liu et al, 2020).

Most proteins that are involved in growth control are degraded by 26S proteasomes; a barrel-shaped degradation complex with one or two lids at the end(s). This degradation pathway usually requires earmarking of the protein to be degraded with a polyubiquitin chain (Sahu & Glickman, 2021). However, ubiquitin-independent proteasome-mediated degradation has also been observed, particularly when proteins to be degraded harbor unstructured regions (Erales & Coffino, 2014). At least some of these ubiquitin-independent degradations occur through alternative lids of the proteasome such as the REG-γ lid (Chen et al, 2007). Particularly in neurons, proteasomes have also been observed to be coupled to microtubules and transported via dynein to their target, providing an alternative mechanism of reaching their target proteins for degradation (Liu et al, 2019). Proteins that are not degraded by proteasomes are frequently degraded by autophagy.

[1]Institute for Biological and Chemical Systems – Biological Information Processing, Karlsruhe, Germany   [2]Human Physiology Department, Medical Research Institute, Alexandria University, Alexandria, Egypt   [3]Division of Oncogenomics, Oncode Institute, Netherlands Cancer Institute, Amsterdam, The Netherlands

Correspondence: christine.blattner@kit.edu; valeria.solozobova@kit.edu
*Seham Elabd and Eleonora Pauletto contributed equally to this work
†Wilbert Zwart and Christine Blattner contributed equally to this work

Here, target proteins are engulfed in a membrane-embraced vesicle, the autophagosome that fuses with lysosomes to form the autolysosome, where proteins are degraded by lysosomal enzymes (Yim & Mizushima, 2020). Other proteins are degraded by proteases that act individually, such as caspases, serine proteases, matrix metalloproteinases, and others (Bond, 2019).

Here, we show that TRIM25 regulates p300 abundance and activity by targeting the transcriptional co-activator for proteasomal degradation. This activity does not require TRIM25-mediated ubiquitination of p300 and is independent of the E3 activity of TRIM25. It also does not involve autophagy although TRIM25 interacts with several proteins of autophagosomes. Instead, TRIM25 is required for the interaction of p300 with the motor protein dynein that forces the migration of cargo along microtubules to the pericentriolar matrix; a cellular structure where proteasomes are enriched. By targeting p300 for degradation, TRIM25 affects gene transcription. This regulation of p300 positions TRIM25 as a putative target for cancer therapy.

# Results

## TRIM25 reduces p300 protein stability

We have previously reported that p300 protein levels were increased in mouse embryonic fibroblasts (MEFs) from mice with a genetic deletion of the *TRIM25* gene (Zhang et al, 2015). In these cells, we furthermore observed increased acetylation of the tumor suppressor protein p53, a known target of p300 (Zhang et al, 2015). These observations implied that TRIM25 may act as a physiologic regulator of p300 abundance and activity, potentially affecting gene transcription.

To investigate this conjecture in more detail, we first confirmed that TRIM25 indeed affects p300 protein levels. We overexpressed HA-tagged *p300* together with *TRIM25* or empty vector for control in H1299 cells and observed a strong decrease in p300 protein levels when *TRIM25* was co-expressed (Fig 1A.I). Endogenous p300 was furthermore decreased in H1299 cells when *TRIM25* was overexpressed

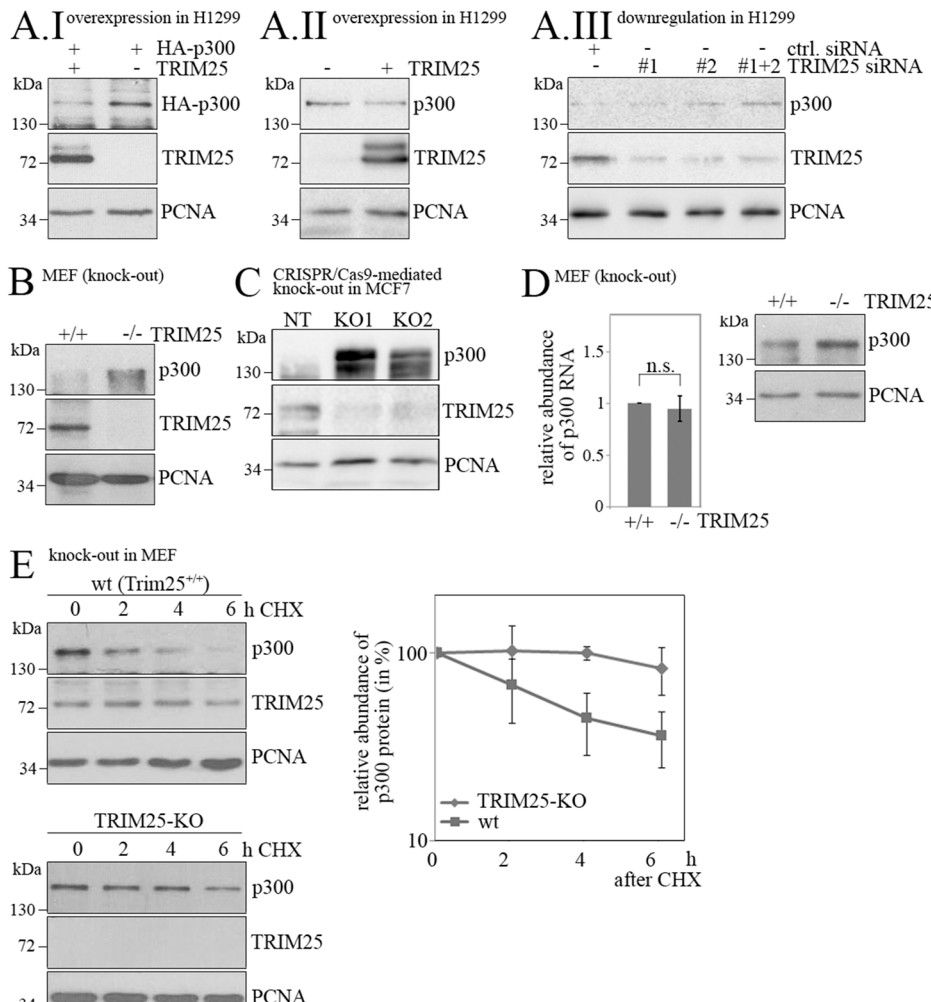

**Figure 1. TRIM25 decreases p300 protein stability.**
**(A.I)** H1299 cells were transfected with HA-*p300* and *TRIM25* as indicated. Transfected p300 and TRIM25 abundance was monitored by Western blotting. Immunodetection of PCNA was performed for loading control. **(A.II)** H1299 cells were transfected with *TRIM25* as indicated. Endogenous p300 and TRIM25 levels were monitored by Western blotting. Immunodetection of PCNA was performed for loading control. **(A.III)** TRIM25 was down-regulated in H1299 cells by siRNA. 40 h after siRNA transfection, levels of p300 and TRIM25 were monitored by Western blotting. Immunodetection of PCNA was performed for loading control. **(B)** MEFs from WT mice (+/+) and from mice with genetically deleted *TRIM25* (−/−) were analysed for p300 and TRIM25 levels by Western blotting. Immunodetection of PCNA was performed for loading control. **(C)** *TRIM25* was knocked-out via CRISPR/Cas9 in MCF7 cells using two different gRNAs (KO1, KO2) and a non-targeting control (NT). Levels of p300 and TRIM25 were monitored by Western blotting. Immunodetection of PCNA was performed for loading control. **(D)** MEFs from WT mice (+/+) and from mice with genetically deleted *TRIM25* (−/−) were harvested. The samples were divided into two parts. One part was used to monitor p300 levels by Western blotting. From the other part, RNA was prepared and the relative abundance of p300 mRNA was monitored by qRT-PCR. Relative abundance of p300 mRNA in WT cells was set to 1. The graph shows mean values and SD of three independent experiments. **(E)** MEFs from WT mice (+/+) and from mice with genetically deleted *TRIM25* (−/−) were treated with 50 µg/ml cycloheximide and harvested at the indicated time points. p300 and TRIM25 abundance was monitored by Western blotting. Immunodetection of PCNA was performed for loading control. The signals for p300 and PCNA were quantified and relative abundance of

p300 was calculated. p300 levels at the time of cycloheximide addition were set to 100. The graph shows mean values and SD of three experiments. Source data are available for this figure.

(Fig 1A.II). Conversely, when TRIM25 was down-regulated by siRNA in H1299 cells, p300 protein levels were increased (Fig 1A.III). As shown before (Zhang et al, 2015), p300 levels were elevated in MEFs from mice where *TRIM25* was genetically deleted in comparison with the corresponding wild-type (WT) cells (Fig 1B). Also, when *TRIM25* was knocked-out by CRISPR/Cas9 or knocked-down by shRNA in MCF7 cells, p300 protein levels were increased (Figs 1C and S1A). This regulation of p300 by TRIM25 occurred purely at the protein level: The half-life of p300 was strongly increased in MEFs with a genetic deletion of the *TRIM25* gene, whereas no difference was observed in p300 mRNA levels of MEFs from *TRIM25* knock-out and WT mice (Fig 1D and E). These results from different cell line models, generated from different tissues, show that TRIM25 regulates p300 in a context-independent manner.

The decrease in protein levels upon *TRIM25* overexpression was specific to p300 and was not seen for other histone acetyltransferases like PCAF (p300/CREB-binding protein-associated factor) or MOZ (monocytic leukemic zinc finger; Fig S1B.I and B.II). Contrarily, increased levels of TIP60 were observed when *TRIM25* was overexpressed (Fig S1B.III). Of note, even CBP, which is highly homologous to p300, was not increased when *TRIM25* was knocked out (Fig S1B.IV). Jointly, these data illustrate selective targeting and degradation of p300 by TRIM25.

## TRIM25-mediated p300 degradation does not require ubiquitination

TRIM25 mediates selective p300 degradation. Because TRIM25 is an E3 enzyme, we first speculated that TRIM25 may ubiquitinate p300, and by this, target it for proteasomal degradation. However, when we performed ubiquitination assays, we could not detect increased ubiquitination of p300 upon *TRIM25* overexpression (Fig 2A). Likewise, we could not detect ubiquitination of p300 when we aimed to detect p300 with Tandem Ubiquitin Binding Entities (TUBES [Hjerpe et al, 2009] Fig 2B). To confirm that TRIM25 does not mediate p300 ubiquitination, we performed proximity ligation assays (PLA), aimed to detect direct interactions between TRIM25 and ubiquitin. Although PLA signal was observed, indicative of p300 ubiquitination, this signal did not change when *TRIM25* was knocked out (Fig S2). Also, when we inhibited the proteasome by treating the cells with epoxomicin, PLA signal—indicative of p300 ubiquitination—was not enhanced. Moreover, the PLA signal for p300 and ubiquitin was much weaker than the PLA signal for ubiquitin and the tumor suppressor protein p53, a well-known target for ubiquitination (Fig S2; Pan & Blattner, 2021).

A prerequisite for ubiquitination is a functional RING domain of the E3 enzyme. To determine whether the E3 ligase activity of TRIM25 is involved in mediating p300 degradation, we mutated two essential cysteines of the TRIM25 RING into an alanine (C30A/C33A). In line with the inability of TRIM25 to ubiquitinate p300, the TRIM25 C30A/C33A mutant was still capable of down-regulating p300, suggesting that TRIM25 may rather serve as a scaffold for mediating p300 degradation, irrespective of its E3 ligase capacity (Fig 2C). In summary, these experiments show that TRIM25 does not mediate p300 ubiquitination. TRIM25 did however co-precipitate with p300, indicating that the two proteins interact (Fig 2D).

## TRIM25 enhances the interaction of p300 with dynein

As TRIM25 does not ubiquitinate p300, we hypothesized that TRIM25 might target p300 in a ubiquitin-independent fashion for degradation. Several TRIM proteins have been shown to function as autophagy adapters (Mandell et al, 2014) and we have also observed an interaction of TRIM25 with GABA-L1, GABA-L2, GABARAB, and LC3, which are key components of autophagosomes (Fig S3A–F). We therefore speculated that TRIM25 might target p300 for degradation by autophagy. To test this hypothesis, we transfected H1299 cells with *p300* and *TRIM25*, treated the cells with the autophagy inhibitor bafilomycin A, and monitored p300 protein levels. We observed a very small decrease of p300 levels upon bafilomycin A treatment, but the decrease in p300 levels after TRIM25 co-transfection was not affected. At the same time, we observed a clear increase in LC3 protein levels, showing that autophagy was indeed inhibited under these conditions (Fig 3A). These results were confirmed when we inhibited autophagy with ammonium chloride (Fig S3G).

Neither autophagy nor TRIM25-mediated ubiquitination of p300 could explain TRIM25's capacity to target p300 for degradation. Although degradation by cellular proteasomes usually requires tagging of the target protein with ubiquitin, ubiquitin-independent degradation has also been observed, particularly for proteins that retain unstructured regions (Lilienbaum, 2013; Ben-Nissan & Sharon, 2014). p300 contains unstructured regions (Kirilyuk et al, 2012), rendering it a possible candidate for this ubiquitin-independent mode of protein degradation by cellular proteasomes. To test this hypothesis, we first assessed whether the TRIM25-mediated degradation of p300 requires functional proteasomes. Therefore, we inhibited proteasomes by treating the cells with epoxomicin. Indeed, treating MCF7 cells with 7.5 $\mu$M epoxomicin for 24 h increased p300 protein levels and abolished the difference in the amount of p300 protein between non-targeted and TRIM25 knock-out MCF-7 cells (Fig 3B), confirming earlier studies that reported p300 as a target of cellular proteasomes (Poizat et al, 2000). To further solidify these observations, we treated MCF7 cells with the proteasome inhibitor epoxomicin and monitored p300 protein half-life. We observed a significant stabilization of the p300 protein when the proteasome was inhibited (Fig S4A). To investigate whether this stabilization was, at least in part, driven by inhibition of TRIM25-mediated degradation, we transfected *p300* in H1299 cells either alone or together with *TRIM25*, and treated the cells with the proteasome inhibitor epoxomicin or with the vehicle DMSO for control. When only *p300* was transfected, the p300 protein was almost completely stable and the stability was not significantly changed by treating the cells with epoxomicin (Fig S4B.I). In contrast, co-transfection of *p300* with *TRIM25*, reduced the half-life of the transfected p300 to about 2 h (Fig S4B.II). Importantly, treating cells with the proteasome inhibitor fully stabilized p300 even when TRIM25 was co-transfected, further confirming that TRIM25 targets p300 for proteasomal degradation (Fig S4B.II).

Because p300 is not ubiquitinated by TRIM25 (Figs 2A and B and S2), the question arose how p300 could be targeted to the proteasome. One possibility was that an adapter protein might connect p300 and the proteasome in a TRIM25-dependent manner. A protein known for such an adapter activity is p62-SQMT, which

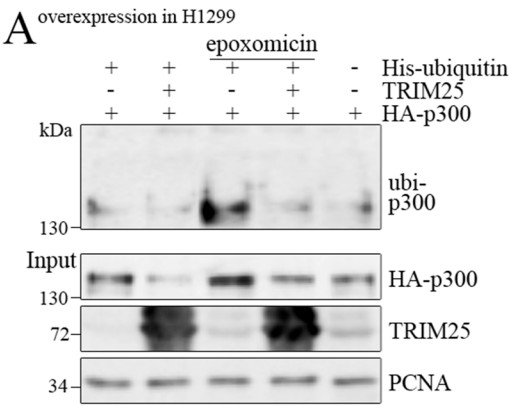

A overexpression in H1299

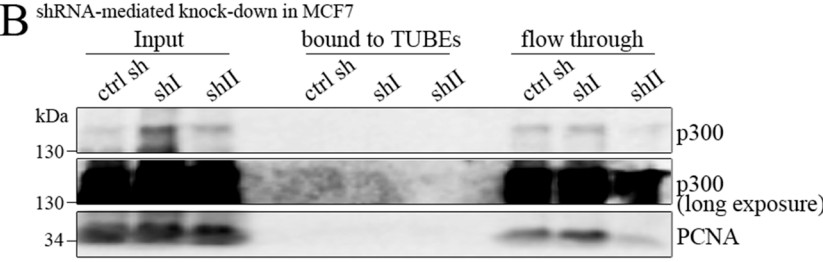

B shRNA-mediated knock-down in MCF7

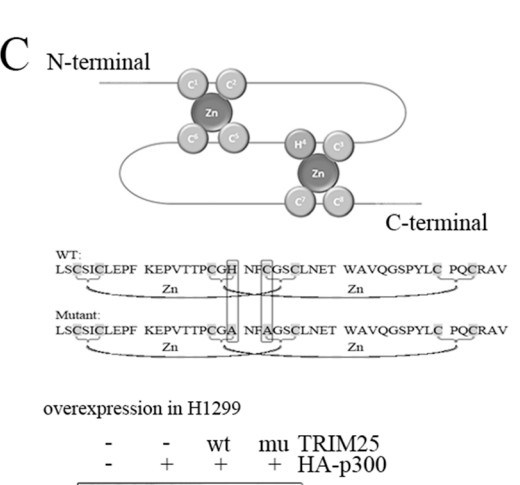

C N-terminal

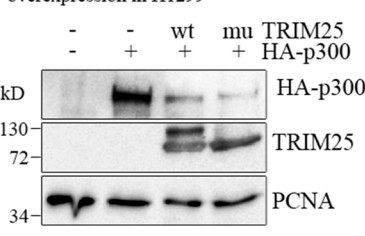

overexpression in H1299

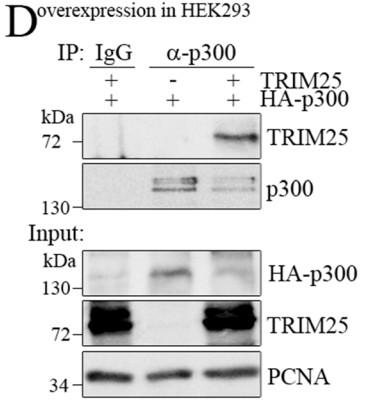

D overexpression in HEK293

**Figure 2. TRIM25 does not mediate p300 ubiquitination.**

**(A)** H1299 cells were transfected with plasmids encoding HA-tagged *p300*, *TRIM25*, and His-tagged *ubiquitin* or with vector DNA for control. 24 h after transfection, cells were treated with 5 μM epoxomicin were indicated. 16 h after epoxomicin addition, cells were harvested and divided into two aliquots. One of the aliquots was used to monitor the abundance of p300 and TRIM25 in the total cell lysate by Western blotting. From the second aliquot, ubiquitinated proteins were purified by adsorption to Ni2+agarose and separated by SDS–PAGE. p300 was monitored by Western blotting. **(B)** MCF7 cells that had been infected with lentiviruses carrying shRNAs targeting TRIM25 (shI, shII) or with a control shRNA (ctrl sh) were lysed. 1/10 of the volume was taken for input. The rest of the lysate was used to precipitate ubiquitinated proteins with TUBEs (Tandem Ubiquitin Binding Entities). TUBEs were collected by centrifugation, washed and bound proteins were eluted. 10% of the flow through was saved for control. All fractions were assessed for p300 by Western blotting. Immunodetection of PCNA was performed for loading control. **(C)** Schematic drawing of the RING domain of TRIM25 and the introduced mutations. H1299 cells were transfected with HA-tagged WT *p300* together with WT *TRIM25* or a *TRIM25* mutant (C30A/C33A). 48 h after transfection, cells were harvested. HA-p300 and TRIM25 levels were monitored by Western blotting. Immunodetection of PCNA was performed for loading control. **(D)** HEK293T cells were transfected with HA-tagged *p300* and with *TRIM25* as indicated. 48 h after transfection, cells were lysed in RIPA buffer. 50 μg of the lysate were used to monitor p300 and TRIM25 levels by Western blotting. Immunodetection of PCNA was performed for loading control. 500 μg of the remaining lysate were used to precipitate p300. Precipitation with IgG was performed for control. p300 and associated TRIM25 were monitored by Western blotting.

Source data are available for this figure.

interacts with the proteasome via its PB-1-domain (Geetha et al, 2008). However, p300 did not co-immunoprecipitate with p62, whereas a weak interaction of p62 with TRIM25 was observed (Fig 3C). Because p300 was previously found in aggresomes upon proteasome inhibition (Kirilyuk et al, 2012), we next tested whether TRIM25 drives p300 into aggresomes to target it for degradation. Aggresomes are microtubule-dependent cytoplasmic inclusion bodies that include subunits of the proteasome and intermediate filaments. These inclusion bodies are formed to sequester

misfolded proteins and cytoplasmic protein aggregates and to deliver them for degradation (Johnston et al, 1998). Importantly, although subunits of the proteasome are associated with aggresomes, ubiquitination of the targets is not a prerequisite for their destruction (Johnston et al, 1998; Kopito, 2000). To test whether TRIM25 drives p300 into higher-order protein complexes, indicative of aggresomal inclusion, we performed sucrose gradient centrifugation of MCF7 cells with or without knocking out *TRIM25* by CRISPR/Cas9. However, p300 was only found as a single molecule or

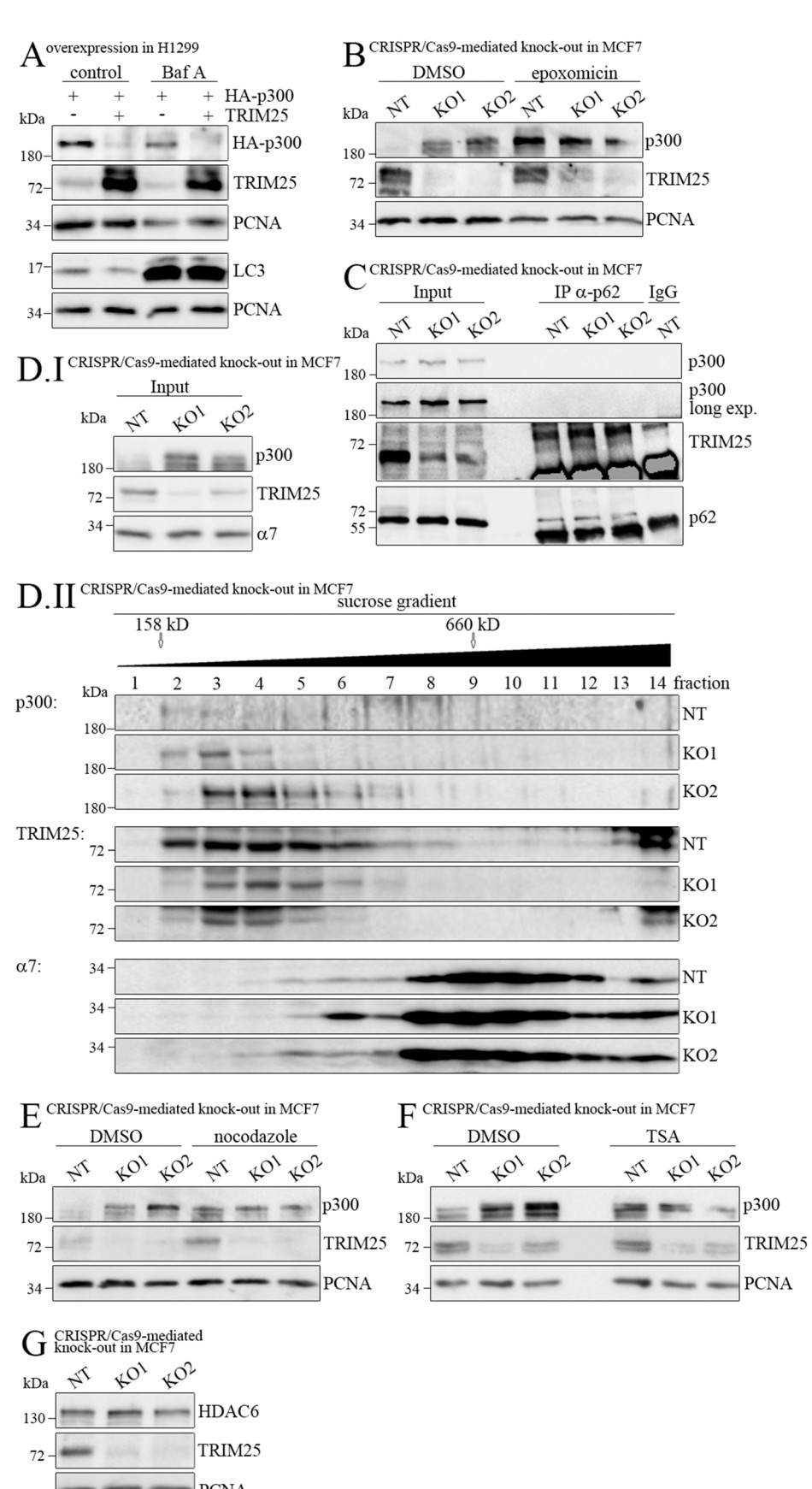

**Figure 3.  TRIM25-mediated p300 degradation requires active proteasomes.**

**(A)** H1299 cells were transfected with HA-tagged WT *p300* together with WT *TRIM25* as indicated. 24 h after transfection, bafilomycin A (Baf A; 200 nM f.c.) was added. 20 h after bafilomycin A addition, cells were harvested and HA-tagged p300, TRIM25, and LC3 were monitored by Western blotting. Immunodetection of PCNA was performed for loading control. **(B, C, D, E, F, G)** MCF7 cells where *TRIM25* was knockout cells with two different gRNAs (KO1, KO2) and a non-targeting control gRNA (NT) were employed. For control, non-targeted mock-treated MCF7 cells were used (NT). **(B)** Cells were treated with 7.5 μM epoxomicin or with DMSO for 24 h. Cells were lysed and p300 and TRIM25 levels were monitored by Western blotting. Immunodetection of PCNA was performed for loading control. **(C)** Cells were cross-linked for 5 min with 0.5% formaldehyde and neutralized with glycine. The samples were lysed and an aliquot was used to monitor p300, TRIM25, and p62-SQMT levels (Input). From 2 mg of the remaining lysate, p62-SQMT was precipitated and associated p300 and TRIM25 levels were monitored by Western blotting. **(D)** Cells were lysed. An aliquot of the lysate was used to monitor p300 and TRIM25 levels. Immunodetection of the proteasomal protein α7 was performed for loading control. 1 mg of the remaining lysate was separated by sucrose gradient centrifugation. The individual fractions were separated by SDS–PAGE. The gels were blotted and p300 and TRIM25 were immunodetected. Immunodetection of α7 was performed to show equal loading and running of the gradients and gels. **(E)** Cells were incubated with nocodazole for 24 h. p300 and TRIM25 levels were monitored by Western blotting. Immunodetection of PCNA was performed for loading control. **(F)** Cells were incubated with trichostatin A for 24 h. p300 and TRIM25 levels were monitored by Western blotting. Immunodetection of PCNA was performed for loading control. **(G)** Cells were lysed. HDAC6 and TRIM25 levels were monitored by Western blotting. Immunodetection of PCNA was performed for loading control.

Source data are available for this figure.

in small protein complexes in the sucrose gradients with no enrichment of p300-containing larger protein conglomerates or aggresomes, irrespective of TRIM25 expression (Fig 3D). Of note, p300 and TRIM25 were present in the same fractions, which is consistent with an interaction of the two proteins (Fig 2D).

Although we could not detect p300 in aggresomes, the association of aggresomes with proteasomal subunits had caught our attention. Because the capacity to transport aggregated proteins via retrograde transport on microtubules appears to be a general mechanism (Kopito, 2000), we hypothesized that TRIM25 uses this cellular machinery to target p300 for degradation. This degradation should be independent of the formation of aggresomes which we could not detect as they mostly form in cells upon proteasome inhibition (Guthrie & Kraemer, 2011; Dehvari et al, 2012; Kirilyuk et al, 2012). However, TRIM25 may use the same cellular components and mechanisms that are present in the cell at all times. If this conjecture is correct, inhibition of microtubule formation should interfere with TRIM25-mediated p300 degradation. To test this possibility, we treated the cells with nocodazole, an inhibitor of tubulin polymerization (Kale et al, 2015). Indeed, incubation of cells with nocodazole strongly increased p300 levels, which did not further increase upon TRIM25 down-regulation (Fig 3E). Microtubules are frequently acetylated at lysine 40 of α-tubulin (Reed et al, 2006; Sadoul & Khochbin, 2016). This acetylation leads to a closer packaging of the globular monomer domain resulting in increased stability and reduced lateral contact points (Sadoul & Khochbin, 2016). Acetylation of microtubules is a reversible process and microtubule deacetylation is performed by SIRT2 (sirtuin type 2 deacetylase) or HDAC6 (histone deacetylase 6; Nekooki-Machida & Hagiwara, 2020). Therefore, we treated cells with the HDAC6 inhibitor trichostatin A, which completely blocked TRIM25-mediated degradation of p300 and no further increase in p300 levels was observed after TRIM25 down-regulation (Fig 3F). The same tendency was observed when cells were treated with the pan-HDAC inhibitor suberoylanilide hydroxamic acid (SAHA), although the effect was less pronounced (Fig S5A). Because HDAC6 activity was required for p300 degradation, TRIM25 could control p300 degradation by regulating HDAC protein levels. However, we did not observe differential HDAC6 levels after TRIM25 down-regulation (Fig 3G).

Apart from protein ubiquitination, TRIM25 can also decorate proteins with ISG15 (Zou & Zhang, 2006). This modification enhances the association of proteins with HDAC6 (Nakashima et al, 2015). To test whether TRIM25 modifies p300 with ISG15, we transfected H1299 cells with His-tagged ISG15 and p300 together with TRIM25 or with an empty vector. We purified ISG15-modified proteins by adsorption to $Ni^{2+}$-agarose and monitored p300 levels by Western blotting. However, we could not find evidence for p300 ISGylation by TRIM25, a result that is in line with the observed redundancy of a functional TRIM25 RING domain for p300 degradation (Figs 2C and S5B).

In case TRIM25 would target p300 to proteasomes via retrograde transport, a motor protein would be required. More recently, it was shown that dynein is involved in protein degradation (Li et al, 2013; Yap et al, 2022). Moreover, acetylation of α-tubulin controls the recruitment of dynein to microtubules which can transport cargo to centrosomes where proteasomes are enriched (Wigley et al, 1999; Kopito, 2000; Vallee et al, 2004; Dompierre et al, 2007; Nekooki-

Machida & Hagiwara, 2020). We therefore hypothesized that p300 interacts with dynein in a TRIM25-dependent manner. To test this hypothesis, we precipitated p300 and monitored associated dynein by Western blotting, confirming an interaction of dynein with p300. Moreover, this interaction was not seen when TRIM25 was knocked-out despite increased p300 levels in those cells (Fig 4A). To confirm these observations through orthogonal methods, we performed PLA. Also with this assay, we found a clear interaction of p300 with dynein that was strongly dependent on TRIM25 (Fig 4B).

In summary, we show that TRIM25 promotes p300 degradation in an ubiquitin and ISG15-independent manner by the dynein/HDAC6/microtubule pathway.

### Dynein associates with TRIM25 and with the proteasome

We found that p300 associates with dynein in a TRIM25-dependent manner. We could furthermore show that no E3-activity of TRIM25 is required for targeting p300 for degradation. These findings suggest that TRIM25 has a scaffold function and may connect p300 with dynein. If this conjecture is correct, then TRIM25 should associate with dynein. Indeed, when we precipitated TRIM25 from MCF7 cells, we found dynein associated with TRIM25 (Fig 4C), and vice versa, when we precipitated dynein, TRIM25 co-precipitated (Fig 4D).

The next question was whether dynein is connected to proteasomes. Earlier studies showed that dynein is linked to proteasomes in neurons and that PI31 serves as an adapter that binds to dynein and the proteasome, thus connecting these two proteins (Liu et al, 2020). To see whether dynein also associates with proteasomes in MCF7 cells, we probed the dynein-immunoprecipitation with an antibody against the α7 protein of the proteasome. Indeed, we saw a strong co-precipitation of the α7 protein of the proteasome and dynein under conditions that leave the proteasome intact (Fig 4E). Of note, this interaction of dynein with the proteasome was independent of TRIM25, indicating that TRIM25 only functions as an adapter for p300 and dynein (Fig 4E). To determine whether TRIM25 is required for an interaction of p300 with proteasomes, we performed PLA with antibodies targeting p300 and the α7 protein of the proteasome. Interestingly, p300 clearly interacted with α7 in the presence of TRIM25, but this interaction was strongly reduced when TRIM25 was knocked out (Fig 4F). Previously, it was observed that proteasome inhibition leads to the formation of massive perinuclear aggregates that are rich in proteasomal antigens and that may form "proteolysis centers" (Wojcik et al, 1996). We treated MCF7 cells with epoxomicin and performed PLA to further investigate the interaction of p300 with dynein und with the proteasome. Interestingly, whereas under normal conditions, the foci representing the interaction of p300 with Dynein were relatively small and dispersed throughout the cell, proteasome inhibitor treatment resulted in fewer but larger dots that were clustered around the nucleus (Fig S6A). The strong reduction in the number of foci when TRIM25 was knocked-out was not compensated by proteasome inhibitor treatment (Fig S6A). A similar result was seen when the PLA was performed for p300 and α7 of the proteasome. Again, although the dots were dispersed throughout the cell in the presence of the vehicle, they were clustered around the nucleus in the presence of epoxomicin (Fig S6B).

Together, these results show that TRIM25 is required to bring p300 to cellular proteasomes.

## TRIM25 affects gene transcription

Because p300 is an important transcriptional co-factor and TRIM25 regulates p300 abundance, we speculated that TRIM25 impacts on transcriptional output. To test this, we transfected HEK293T cells with an *MMTV (mouse mammary tumor virus) reporter* that drives luciferase transcription in response to activation of several nuclear receptors including the glucocorticoid receptor (GR) and the androgen receptor. Cells were co-transfected with the p300-dependent *GR* (Conway-Campbell et al, 2011) along with *TRIM25* or empty vector control and treated with dexamethasone, after which, relative reporter activity was measured. Interestingly, dexamethasone-induced GR activity was strongly reduced upon TRIM25 co-expression, whereas GR protein levels remained unaltered (Fig 5A.I). Contrarily, knocking out *TRIM25* in MEFs strongly increased GR activity, both in the absence and presence of dexamethasone (Fig 5A.II). To test for generalizability of our observations, we next determined the impact of TRIM25 levels on the activity of a second p300-dependent transcription factor: the androgen receptor (AR; Waddell et al, 2021). We transfected HEK293 cells with the *MMTV luciferase reporter* together with *AR* and *TRIM25* or empty vector control and treated the cells with dihydrotestosterone. In agreement with our prior observations, TRIM25 overexpression reduced both basal and hormone-induced androgen receptor activity (Fig 5B.I), whereas knocking out TRIM25 increased AR activity (Fig 5B.II). To

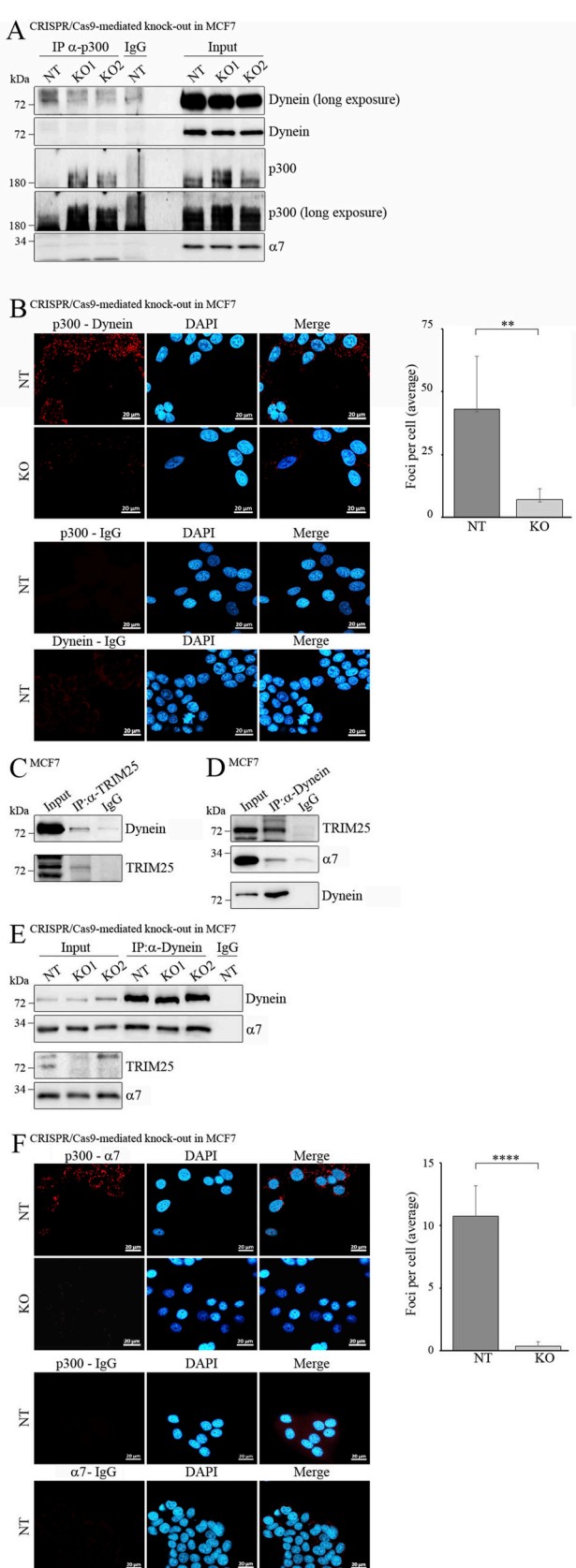

**Figure 4. Dynein associates with TRIM25 and with the proteasome.**
**(A)** MCF7 cells where *TRIM25* was knockout cells with 2 different gRNAs (KO1, KO2) and a non-targeting control gRNA (NT) were lysed. An aliquot of the lysate was used to monitor p300 and dynein levels. Immunodetection of the proteasomal protein α7 was performed for loading control. From 3.2 mg of the remaining lysate, p300 was precipitated, and associated Dynein was monitored by Western blotting. **(B)** MCF7 cells where *TRIM25* was knocked out by CRISPR/Cas9 (KO) and a non-targeted control (NT) were plated in chamber slides. PLA was performed using antibodies against p300 and dynein or against p300 and mouse IgG or against dynein and rabbit IgG. Nuclei were visualized by DAPI (*blue*), the PLA signals were labelled with Texas red (*red*). Images were obtained using a Zeiss LSM-900 confocal microscope. The dots of 78 non-targeted cells and of 87 knock-out cells were counted. Mean values and standard deviations were plotted. (\*\*: P < 0.01). **(C)** MCF7 cells were lysed. An aliquot of the lysate was used to monitor dynein and TRIM25 levels. From 5 mg of the lysate, TRIM25 was precipitated. Two-thirds of the precipitate were loaded onto one gel, blotted, and associated dynein was monitored by Western blotting. The remaining third was loaded onto a second gel, blotted, and probed for TRIM25. **(D)** MCF7 cells were lysed. An aliquot of the lysate was used to monitor dynein, α7, and TRIM25 levels. From 5 mg of the lysate, dynein was precipitated. Two-thirds of the precipitate were loaded onto one gel, blotted, and associated TRIM25 and α7 were monitored by Western blotting. The remaining third was loaded onto a second gel, blotted, and probed for dynein. **(E)** Non-targeted and TRIM25 Knock-out MCF7 cells were lysed. Aliquots of the lysate were used to monitor dynein and α7 and TRIM25 levels. Immunodetection of α7 was performed for loading control. From 3.8 mg of the lysates, dynein was precipitated. Associated α7 was monitored by Western blotting. **(F)** MCF7 cells where *TRIM25* was knocked-out by CRISPR/Cas9 (KO) and a non-targeted control (NT) were plated in chamber slides. PLA was performed using antibodies against p300 and α7 or against p300 and mouse IgG. Nuclei were visualized by DAPI (*blue*) and the PLA signals were labelled with Texas red (*red*). Images were obtained using a Zeiss LSM-900 confocal microscope. The dots of 200 non-targeted cells and of 236 knock-out cells were counted. Mean values and standard deviations were plotted. (\*\*\*\*: P < 0.001).
Source data are available for this figure.

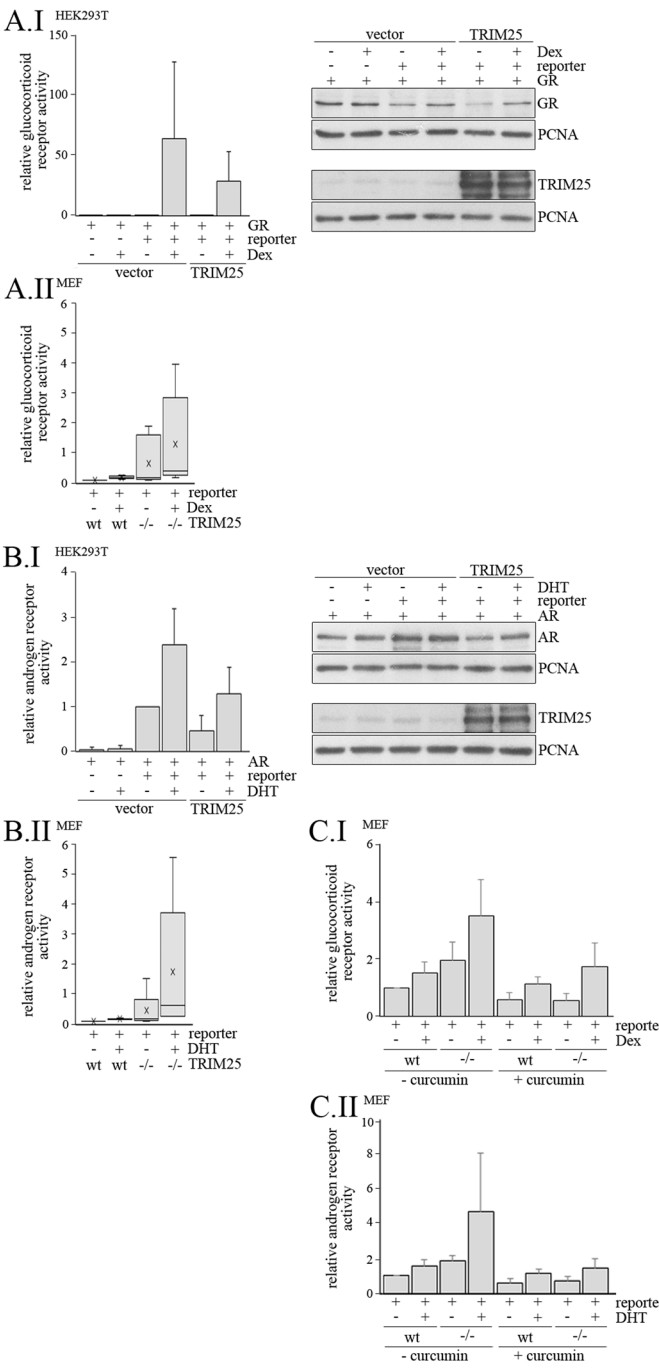

**Figure 5. TRIM25 regulates the activity of the androgen and of the glucocorticoid receptor.**

**(A.I)** HEK293T cells were transfected in triplicates with plasmids harboring the *glucocorticoid receptor (GR)*, a *pGL3-MMTV firefly luciferase* and a *renilla luciferase* reporter, for internal control. Where indicated, a plasmid encoding *TRIM25* or vector DNA was co-transfected. 42 h after transfection, cells were treated with $10^{-7}$ M dexamethasone (Dex) or with vehicle. 6 h after hormone addition, cells were harvested. One of the triplicates was used to monitor abundance of the glucocorticoid receptor (GR), and TRIM25 by Western blotting. Immunodetection of PCNA was performed for loading control. The remaining two triplicates were analysed for firefly and Renilla luciferase activity. Relative firefly activity was calculated and firefly activity of cells transfected with GR, luciferase, and Renilla reporter and treated with vehicle was set to 1. The graph shows mean values and standard deviations of three independent experiments. **(A.II)** WT and *TRIM25*$^{-/-}$

confirm functional dependency of p300 for both AR and GR activity, the cells were treated with the p300 inhibitor curcumin (Morimoto et al, 2008). Whereas the effect of curcumin on androgen and GR activity was less prominent in WT MEFs, the activity was strongly reduced in *TRIM25* knock-out cells, particularly in the presence of hormones (Fig 5C). These data jointly illustrate the impact of TRIM25 on p300-driven gene expression.

To systematically determine the impact of TRIM25 on p300-depedent gene expression in yet another context, we performed RNA-seq in the estrogen receptor α-dependent breast cancer cell line MCF7, in which *TRIM25* was knocked-out by CRISPR/Cas9 (Fig S7A). Principle component analyses (PCAs) showed clear clustering of the non-targeted replicates. TRIM25 knock-out replicates located away from the non-targeted controls, but only two of the three replicates were grouped (Fig S7B). We therefore decided to further analyze only the grouped KO replicates and also only two of the non-targeted replicates. The heatmap shows high correlation between samples ($R^2 > 0.9$; Fig S7C).

To selectively target our analyses on TRIM25-driven p300 modulation, we integrated the RNA-seq dataset with p300 ChIP-seq data in MCF7 cells, generated under a vehicle or a 3-h estrogen

MEFs were transfected in duplicates with plasmids harbouring a *pGL3-MMTV firefly luciferase* and a *renilla luciferase* reporter for internal control. 42 h after transfection, cells were treated with $10^{-7}$ M Dex or with vehicle. 6 h after hormone addition, cells were harvested and analysed for firefly and Renilla luciferase activity. Relative glucocorticoid receptor activity of wt MEFs transfected with the reporter and treated with vehicle was set to 1. The box plot shows mean values and SD of relative glucocorticoid receptor activity of seven independent experiments. **(B.I)** HEK293T cells were transfected in triplicates with plasmids harboring the *androgen receptor* (AR), a *pGL3-MMTV firefly luciferase* and a *renilla luciferase* reporter, for internal control, and where indicated, with a plasmid encoding *TRIM25* or with vector DNA. 24 h after transfection, cells were treated with $10^{-8}$ M dihydroxytestosteron (DHT) or with vehicle. 24 h after hormone addition, cells were harvested. One of the triplicates was used to monitor abundance of the androgen receptor (AR) and TRIM25 by Western blotting. Immunodetection of PCNA was performed for loading control. The remaining two triplicates were analyzed for firefly and luciferase activity and relative firefly activity was calculated. The graph shows mean values of relative androgen receptor activity of three independent experiments. Relative androgen receptor activity of HEK cells transfected with *AR*, *firefly luciferase*, and *renilla luciferase* reporter and treated with vehicle was set to 1. **(B.II)** WT MEFs and MEFs where *TRIM25* has been genetically deleted (−/−) were transfected in duplicates with plasmids harboring a *pGL3-MMTV firefly luciferase* and a *renilla luciferase* reporter for internal control. 24 h after transfection, cells were treated with $10^{-8}$ M DHT or with vehicle. 24 h after hormone addition, cells were harvested and analyzed for firefly and renilla luciferase activity. Relative firefly activity was calculated. The box plot shows mean values of relative androgen receptor activity of six independent experiments. Relative androgen receptor activity of WT MEFs transfected with the *firefly luciferase* reporter and treated with vehicle was set to 1. **(C)** WT and TRIM25$^{-/-}$ MEFs were transfected in duplicates with plasmids harbouring a *pGL3-MMTV firefly luciferase* and a *renilla luciferase* reporter, for internal control. **(C.I)** 39 h after transfection, curcumin was added to a final concentration of 40 μM. 3 h after curcumin addition, cells were treated with $10^{-7}$ M Dex or with vehicle. 6 h after hormone addition, cells were harvested and analysed for firefly and Renilla luciferase activity. Relative firefly activity was calculated and firefly activity of WT cells treated with vehicle was set to 1. The graph shows mean values and standard deviations of four independent experiments. **(C.II)** 21 h after transfection, curcumin was added to a final concentration of 40 μM. 3 h after curcumin addition, cells were treated with $10^{-8}$ M DHT or with vehicle. 24 h after DHT addition, cells were harvested and analysed for firefly and renilla luciferase activity. Relative firefly activity was calculated and firefly activity of WT cells treated with vehicle was set to 1. The graph shows mean values and standard deviations of four independent experiments. Source data are available for this figure.

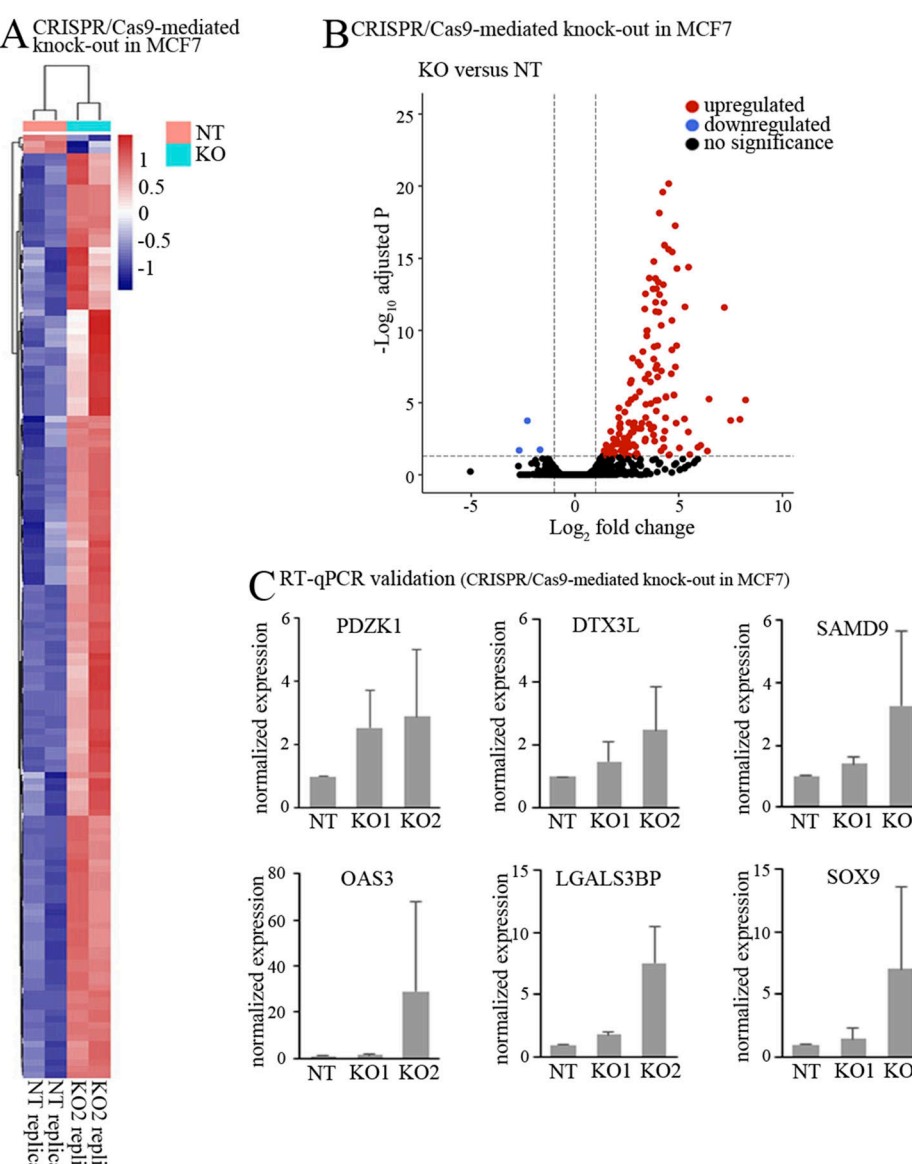

**Figure 6. Down-regulation of TRIM25 changes gene expression.**
**(A)** Heatmap representing normalised values derived from variance stabilising transformation (according to DeSeq2 package), filtered for close proximity (20 kb from transcription start site) to confirmed p300 binding sites, comparing TRIM25-KO versus NT control. Red colour indicates genes with high expression levels, and blue colour indicates genes with low expression levels. Two independent experiments were done for each cell line. **(B)** Volcano plot of the p300 binding sites showing the differentially expressed genes when comparing knockdown of TRIM25 to non-targeted control. Cut-offs are fold change >1 or <−1 and adjusted $P$-value > 0.05. (black: not significant, red dot: up-regulated in NT; blue: down-regulated in KO). Two independent experiments were done for each cell line. **(C)** RNA was prepared from MCF7 cells with CRISPR-Cas9–mediated knock-out of *TRIM25* (KO1, KO2) or treated with a non-targeted control (NT). Relative abundance of PDZK1, DTX3L, SAMD9, OAS3, LGAL53BP, and SOX9 mRNA was monitored by qRT–PCR. Relative abundance of these mRNAs in the non-targeted control cell line was set to 1. The graphs show mean values and SD of three independent experiments.

treatment (Zwart et al, 2011). We selected 10,360 peaks from the overlap between vehicle and estrogen-treated cells (corresponding to estrogen-independent p300 binding) and 9,677 peaks present only with estrogen treatment (corresponding to estrogen-induced p300 binding). We associated each peak to the nearest transcription start site of the genes within a range of 20 kb. By this, we obtained a list of 4,402 putative direct p300 target genes. Out of these genes, 148 genes were significantly up-regulated when TRIM25 was knocked-out, whereas only three genes were down-regulated as shown in the heatmap analyses and volcano plot (Fig 6A and B). RNA-seq results were successfully and independently validated by qRT–PCR using two independent MCF7 TRIM25 knock-out lines (the reduction in TRIM25 is shown in Fig S7D), confirming up-regulation of PDZK1, DTX3L, SAMD9, OAS3, LGAL53BP or SOX9 when *TRIM25* was knocked out (Fig 6C). Jointly, these data position TRIM25 as a key

regulator of p300 levels, impacting the activity of several critical transcription factors.

## Discussion

Proteasome-dependent degradation is classically controlled by ubiquitination of the target protein, a reaction catalyzed by E3 proteins. Once the target protein carries more than 4 ubiquitin residues in a chain, it is recognized by the proteasome, engulfed, and degraded (Lilienbaum, 2013). Most RING (really interesting new gene)-containing proteins have E3 activity and can mediate the transfer of ubiquitin to target proteins, priming them for degradation (Pan & Blattner, 2021). One of the common characteristics of TRIM

protein family members is the N-terminal RING domain and indeed, TRIM25 mediates the transfer of ubiquitin (and also of its closely related homologue ISG15) onto target proteins to target them for proteasomal degradation (Urano et al, 2002; Zou & Zhang, 2006).

Here, we report that degradation of p300 by TRIM25 does not use this well-investigated ubiquitin-dependent route for protein degradation, whereas TRIM25–p300 protein interactions were detected. Of note, ubiquitin-independent degradation, although occurring only rarely, has been previously reported with proteins with an inherent structural disorder (Lilienbaum, 2013; Ben-Nissan & Sharon, 2014) as prime examples. A disordered region has also been identified in p300 (Kirilyuk et al, 2012), making it likely that p300 might also be able to use an ubiquitin-independent route into the proteasome.

The question was then, how TRIM25 could bring p300 to the proteasome. Autophagy, p62-dependent delivery to proteasomes or the formation of aggresomes seemed not to be involved in TRIM25-driven degradation of p300. However, if there is a pathway that sequesters p300 in aggresomes when proteasomes are inhibited, then this pathway should allow p300 degradation under normal conditions, when proteasomes are active. Because aggresomes assemble by retrograde transport of, for example, misfolded proteins towards the minus end of microtubule and their formation are blocked by drugs that depolymerize microtubules; nocodazole should block TRIM25-mediated degradation of p300 if this pathway is employed in TRIM25-mediated p300 degradation. Indeed, treatment of cells with nocodazole blocked TRIM25-mediated degradation of p300.

Intracellular delivery over distances is usually driven by motor proteins that ferry cargo along microtubule tracks. Kinesin and dynein are the most well-known and most investigated motor proteins. These two proteins transport cargo either to the plus end in the periphery (kinesins) or to the minus end in the cell center (dynein) (Hirokawa & Takemura, 2004; Vallee et al, 2004). Of note, proteasomes have been found at centrosomes and also proteasome-mediated proteolysis associated with centrosomes has been observed (Wigley et al, 1999; Kopito, 2000), suggesting that among these motor proteins, dynein might be the one being involved in p300 degradation. Polarized trafficking furthermore involves specific posttranslational modifications. Acetylation of α-tubulin at lysine 40, for instance, enhances the binding of kinesin to microtubules (Reed et al, 2006). To further support the hypothesis that dynein microtubule-dependent transport to proteasomes drives TRIM25-mediated p300 degradation, we treated cells with the HDAC inhibitors trichostatin A and SAHA. Trichostatin A blocked TRIM25-mediated p300 degradation completely and also SAHA interfered with this process, showing that deacetylation of proteins is an important intermediate for p300 degradation. Interestingly, HDAC6 is not only a tubulin deacetylase (Hubbert et al, 2002) but also links cargos of aggregated proteins to dynein (Kopito, 2003). Moreover, p300 has been shown to interact specifically with HDAC6 and also with dynein (Kirilyuk et al, 2012). We hypothesized that the E3 enzyme TRIM25 could affect HDAC levels but this was not the case. Yet, we did observe a strong decrease in p300 binding to dynein when TRIM25 was knocked out. We thus conclude that TRIM25 is required for an interaction of p300 with dynein which allows the transport of p300 in a HDAC-dependent manner along

microtubules to pericentriolar proteasomes where p300 is degraded in an ubiquitin-independent manner (Fig 7). This interaction of dynein with p300 is mediated by a scaffold function of TRIM25 that co-precipitated with dynein. Dynein, on the other hand, co-precipitated with the proteasome which allows the carrying of its cargo, the p300 protein, to the proteasome, whereas the requirement of modifying p300 with ISG15, a modification that has been shown to enhance the interaction of other proteins with HDAC6 (Nakashima et al, 2015) can be excluded.

The regulation of p300 by TRIM25 translates into the regulation of transcription. Large-scale transcriptional alterations of p300-driven genes were observed upon TRIM25 perturbation, although the level of regulation was in most cases modest. Consistent with a rate-limiting role of transcriptional co-activators, most of these genes were up-regulated when TRIM25 levels were reduced (and therefore p300 levels were increased). The regulation of p300 by TRIM25 appears to be largely specific as we could not see a similar down-regulation of other histone acetylases like PCAF, CBP or MOZ (monocytic leukemia zinc finger protein) by TRIM25. We found, however, higher levels of TIP60 in WT MEFs in comparison with MEFs where TRIM25 was genetically deleted. Of note, an interaction with dynein has not been found for CBP (Kirilyuk et al, 2012), which explains why CBP is not regulated by TRIM25 despite the high level of homology between p300 and CBP.

p300 has been found to be overexpressed in cancer cells where it contributes to the activation of oncogene transcription and cell proliferation. Conversely, small molecule inhibitors of p300 were found to down-regulate oncogene transcription and cancer cell proliferation and enhance the anticancer effect of chemotherapy and radiation therapy in mouse models. Two of these inhibitors are currently in clinical trials (Chen et al, 2022). Targeting p300 for degradation puts TRIM25 in the heart of p300 regulation, which makes TRIM25 an interesting target for cancer therapy.

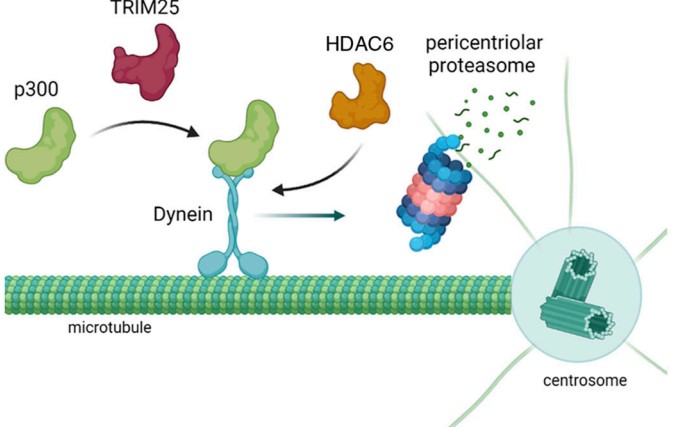

**Figure 7. TRIM25 targets p300 for degradation in a dynein-dependent manner.**
TRIM25 is required for the interaction of p300 with dynein that transports it in a HDAC-dependent manner along microtubules to (pericentriolar) proteasomes (schematic drawing created with BioRender.com).

# Materials and Methods

## Antibodies and reagents

Most of the antibodies were from Santa Cruz (anti-proliferating cell nuclear antigen [PC10, sc-56], anti-Efp [E-12, sc-271254 and E4, sc-166926], anti-p300 for human p300 [NM11, sc-32244], anti-p62 [D3, sc-28359], anti-Dynein [74-1, sc13524], anti-GR [P20, sc-1002], anti-CBP [C-1, sc-7300], anti-PCAF [H369, sc8999], anti-MOZ [4D8, sc-293283], anti-p53 [FL393, sc-6243], anti-GST [B-14, sc-138], and the anti-androgen receptor antibody [441, sc-7305]). Other antibodies (anti-LC3 [D3U4C, #12741P], anti-HDAC6 [D21B10, #7612S], anti-p300 [D8Z4E, #86377], anti-ISG15 [#2743S]) were from Cell Signaling. The anti-p300 antibody for murine p300 (RW128, 05-257) was from Millipore, the anti-HA high-affinity antibody for HA-tagged p300 (#11867423001) from Roche, the anti-TIP60 antibody (#07-038) was from Upstate Biotechnology, and the anti-$\alpha$7 antibody (MCP72, BML-PW8110-0100) and the anti-ubiquitin antibody (FK2, PW8810-0500) were from Enzo.

Cells were treated with the following inhibitors from Sigma-Aldrich: cycloheximide (C1988) dissolved in $H_2O$ (f.c. 50 mg/ml), nocodazole (N1404) dissolved in DMSO (f.c. 20 $\mu$M), trichostatin A (T8552) dissolved in DMSO (f.c. 500 nM), dexamethasone (D8893) dissolved in ethanol (f.c. $10^{-7}$M), and dihydrotestosterone (R189286) dissolved in ethanol (f.c. $10^{-8}$M). Epoxomicin (BML-PI127) was from Enzo (f.c. 5 $\mu$M and 7.5 $\mu$M) and was dissolved in DMSO. Autophagy was inhibited with 50 mM (f.c.) NH4Cl (#K298.2) from Karl Roth, dissolved in $H_2O$ and with bafilomycin A (sc-201550) from Santa Cruz (f.c. 200 nM), dissolved in DMSO. Curcumin (8.20254) was from Merck Millipore (f.c. 40 $\mu$M) and SAHA (#100009929) from Cayman. Both were dissolved in DMSO.

peqGOLD TriFast was purchased from VWR. Enzymes for cDNA synthesis were purchased from Promega and the Takyon 2x Rox SYBR Mastermix blue dTTP was obtained from Eurogentec.

TUBEs (Hjerpe et al, 2009) were kindly provided by Manuel Rodriguez, Toulouse, France.

## Cell lines and their treatments

PC3 cells were cultured in Roswell Park Memorial Institute 1640 medium containing 10% FBS 1% L-glutamine and 1% penicillin/streptomycin. H1299, MCF7, MEFs, and HEK 293T cells were cultured in DMEM containing 10% FBS and 1% penicillin/streptomycin at 6% $CO_2$. All cells were kept at 37°C in a humidified atmosphere. $TRIM25^{+/+}$ and $TRIM25^{-/-}$ MEFs were given to us by Satoshi Inoue, Tokio, Japan. MCF7 (ctr sh) were generated by infecting MCF7 cells with Mission pLKO.1puro Non-Target shRNA Control Transduction Particles (SHCO16V-1EA), Sigma-Aldrich, according to the protocol from the manufacturer. MCF7 (shTRIM25_I, shTRIM25_II) were generated by infecting MCF7 cells with Mission Transduction Particles targeting TRIM25 (SHCLNV- TRC0000272697 and SHCLNV-TRCN0000438014), Sigma-Aldrich, according to the protocol from the manufacturer. Polyclonal MCF7 non-targeting and knockout cell lines were generated using the lentiCRISPR v2 system (Sanjana et al, 2014). In brief, guides targeting TRIM25 (guide #1 for: CAC-CAAGCACGTCTTCACGG rev: CCGTGAAGACGTGCTTGGTG and guide #2

for: AAAGCCAGTCTACATCCCCG rev: CGGGGATGTAGACTGGCTTT) from the GeCKO library or non-targeting controls were cloned into the lentiCRISPR v2 construct and co-expressed with third generation lentiviral vectors in HEK cells using 105 $\mu$l polyethyleneimine (1 mg/ml). MCF7 cells were infected with the lentiviral particles. 48 h after infection, cells were selected with puromycin (2 $\mu$g/ml f.c) for 2 wk.

HEK293T cells were transiently transfected by calcium–phosphate DNA co-precipitation (Kulikov et al, 2010). H1299 cells were transiently transfected with PromoFectin (PromoKine) or with Lipofectamin 3000 according to the protocols from the manufacturers.

## Small interfering RNA transfection

siRNA transfection was performed in H1299 cells using Lipofect-amine 3000 (#L3000001; Life Technologies, Thermo Fisher Scientific) according to the manufacturer's instructions. Briefly, 2 × $10^5$ cells were transfected with 7.5 $\mu$l of Lipofectamine 3000 and 5 nM or 10 nM TRIM25 siRNA (siRNA-1: 5′-GGGAUGAGUUCGAGUUUCU-3′, siRNA-2: 5′-CUGCGAGGAAUCUCAACAATT-3′) or control siRNA (5′-GGUGCG-CUCCUGGACGUAGCC-3′) in six-well plates. 16 h after transfection, the medium was changed, and after the next 24 h, the cells were harvested for analysis.

## Plasmids

Plasmids encoding the *androgen receptor*, the *glucocorticoid receptor*, and the *MMTV firefly* reporter were provided by Andrew C. Cato, Karlsruhe, Germany. LentiCRISPR v2 was provided by Feng Zhang, Cambridge, MA, USA. HA-tagged *p300* was provided by Steven Grossman, Richmond, VA, USA. *TRIM25* and His-tagged *ubiquitin* were described earlier (Zhang et al, 2015). Plasmids encoding *GABA-L1*, *GABA-L2*, *GABARAB*, *LC3A*, *LC3B*, and *LC3C* were kindly provided by Ivan Dikic, Frankfurt, Germany, the plasmid encoding Flag-tagged *PCAF* was kindly provided by Olivier Coux, Montpellier, France, the plasmid for His-tagged *ISG15* was kindly provided by Gerrit Praefke, Cologne, Germany.

For the *TRIM25–RING mutant* (C30A/C33A), *TRIM25* was amplified by PCR with primers containing the respective mutants and the Q5 Site-Directed Mutagenesis Kit (#E0554; New England Biolabs) according to the manufacturer's protocol. The mutation was verified by sequencing.

Sequences of primers are available on request.

## Western Blotting

The methods of SDS–PAGE and Western blotting are described in Kulikov et al (2010). Proteins were blotted onto nitrocellulose membrane. For some experiments, blotting buffer was supplemented with 0.01% SDS.

## qRT–PCR

Total RNA was prepared from cells using peqGOLD TriFast according to the manufacturer's protocol. The RNA was treated with DNaseI to remove residual genomic DNA and transcribed into cDNA using

random primers and the M-MLV reverse transcriptase. Real-time PCR was performed with a SYBR Green PCR mixture. The cDNA was denatured for 15 min at 95°C followed by 40 cycles of 95°C for 15 s and 50°C for 1 min using the 7,000 ABI sequence detection system and gene-specific primers. The signals were normalized to the signals for the housekeeping genes RibPO (34B4) or actin.

Sequences of primers are available on request.

## Luciferase-reporter assay

For luciferase assays, $1 \times 10^4$ cells per well were plated in 96-well plates. For transfections into HEK293T cells, each well was transfected with 50 ng of the *MMTV-firefly luciferase* reporter and 5 ng of a plasmid encoding *renilla luciferase* together with 25 ng of a plasmid encoding the *glucocorticoid receptor* or the *androgen receptor* or with vector DNA. For transfection into mouse embryonic fibroblasts, each well was transfected with 50 ng of the *MMTV firefly luciferase* reporter and 5 ng of a plasmid encoding *renilla luciferase*. 48 h after transfection, the cells were lysed and the luciferase activity was determined using a Victor Light luminescence counter (Perkin Elmer Inc.).

## Immunoprecipitation

The method of immunoprecipitation is described in Kulikov et al (2006). For the co-immunoprecipitation of p300 and TRIM25, RIPA buffer (50 mM Tris pH 7.5, 150 mM NaCl, 1% Igepal CA-630, 0.5% sodium deoxycholate, 1 mM phenylmethylsulfonyl fluoride) was used to lyse the cells. For the co-immunoprecipitation of p62 and p300, proteins were cross-linked with 0.05% formaldehyde which was quenched with 80 mM (f.c.) glycin. Cells were lysed in 50 mM Tris 7.4, 20 mM NaCl, 10 mM MgCl2. 0.5% Igepal CA-630, and 1 mM phenylmethylsulfonyl fluoride and this buffer was also used for washing the precipitates for five times. For the co-immunoprecipitation of p300, TRIM25, and dynein, cells were lysed in 50 mM Tris 7.4, 20 mM NaCl, 10 mM MgCl2. 0.5% Igepal CA-630, 10% glycerol, and 1 mM phenylmethylsulfonyl fluoride. For the co-immunoprecipitation of the proteasome, ATP was added to a final concentration of 5 mM. This buffer was also used for washing the precipitates five times.

## Tube assay

Cells were trypsinized and counted. Cells were lysed in 500 $\mu$l lysis buffer (50 mM Tris pH 8.5, 150 mM NaCl, 5 mM EDTA, 1% Igepal CA-630, 1 mM PMSF). The lysate was cleared by centrifugation (17,000$g$, 10 min, 4°C). 10% of the cleared lysate was taken for "Input." The remaining lysate was mixed with 100 $\mu$l TUBEs bound to GST beads and incubated for 4 h at 4°C with end-over-end rotation. The beads were collected by centrifugation. The supernatant was saved as "Flow Through." The beads were washed three times with PBS containing 0.05% Tween 20. Proteins were eluted with 40 $\mu$l 2x sample buffer and heated two times for 4 min to 95°C with vortexing in between.

## Sucrose gradient

Cells were washed, scraped in PBS, collected by centrifugation, and lysed for 20 min on ice in lysis buffer (50 mM Tris 7.4, 20 mM NaCl, 10 mM MgCl2. 0.5% Igepal CA-630, 5 mM ATP, 10 mM N-ethylmaleimide, and 1 mM phenylmethylsulfonyl fluoride). For a better lysis, cells were syringed six times with a 26G needle. The protein lysate was cleared by centrifugation 17,000$g$, 15 min, 4°C) and the protein concentration was determined. An equal amount of proteins was loaded onto the top of a 10–40% sucrose gradient (solution 1: 10% sucrose, 25 mM Tris pH 7.4, 50 mM NaCl, 0.05% Igepal CA-630, 1 mM PMSF; solution 2: 40% sucrose, 25 mM Tris pH7.4, 50 mM NaCl, 0.05% Igepal CA-630, 1 mM PMSF). Gradients were centrifuged in an ultracentrifuge at 150,000$g$ for 18 h at 4°C. Fractions were collected and analyzed by Western blotting.

## Ubiquitination assay

H1299 cells were transfected with plasmids encoding HA-tagged *p300*, *TRIM25* and His-tagged *ubiquitin* or with vector DNA for control. 24 h after transfection, cells were treated with 5 $\mu$M epoxomicin and incubated for 16 h. Cells were harvested, washed twice with ice-cold PBS, and lysed in 6 ml of guanidinium lysis buffer pH 8 (6 M guanidinium-HCl, 0.1 M Na$_2$HPO$_4$/NaH$_2$PO$_4$ [pH 8], 0.01 M Tris [pH 8], 5 mM imidazole, 10 mM $\beta$-mercaptoethanol). Ni$^{2+}$-nitrilotriacetic acid-agarose was added and the mixture was incubated by end-over-end rotation at room temperature for 4 h. The agarose was washed once with guanidinium lysis buffer, once with urea buffer pH 8 (8 M urea, 0.1 M Na$_2$HPO$_4$/NaH$_2$PO$_4$ [pH 8], 0.01 M Tris [pH 8], 10 mM $\beta$-mercaptoethanol), once with buffer A pH 6.3 (8 M urea, 0.1 M Na$_2$HPO$_4$/NaH$_2$PO$_4$ [pH 6.3], 0.01 M Tris [pH 6.3], 10 mM $\beta$-mercaptoethanol) and once each with buffer A with 0.2% Triton X-100, and buffer A with 0.1% Triton X-100. Elution was carried out with elution buffer (200 mM imidazole, 5% SDS, 0.15 M Tris [pH 6.7], 30% glycerol, 0.72 M $\beta$-mercaptoethanol). The eluate was diluted 1:1 with 2× SDS–PAGE sample buffer and subjected to SDS–PAGE. The proteins were transferred onto a nitrocellulose membrane and probed with the anti-HA antibody to detect the HA-tagged p300.

## ISGylation assay

The ISGylation assay was essentially done as the ubiquitination assay described above, only that His-tagged *ISG15* was transfected instead of His-tagged ubiquitin.

## GST pull down

Plasmids encoding GST or GST-fused proteins were transfected into bacteria and incubated overnight at 37°C on an ampicillin-containing LB agar plate. The next day, one colony was picked and incubated overnight at 37°C in ampicillin-containing LB medium with constant shaking. The next morning, the bacteria-containing medium was diluted 1:10 with fresh ampicillin-containing LB medium and incubated for 1.5 h at 37°C with constant shaking. IPTG was added (f.c. 1 mM) and incubated for further 8 h. The bacteria were collected by centrifugation (10 min, 5,500$g$, 4°C) and frozen at −80°C. The next day, the pellet was resuspended in 25 ml ice-cold PBS and three times sonicated for 30 s. Between the sonication steps, the lysate was kept on ice. 250 $\mu$l PMSF (100 mM) and 250 $\mu$l Triton-X-100 were added and the bacteria were lysed at 4°C for 40 min with end-over-end rotation. The lysate was cleared twice by centrifugation (first: 10 min, 13,200$g$, 4°C; second:

20 min, 13,200*g*, 4°C). The GST-proteins were purified by adsorption to GST–agarose. Therefore, 100 *μ*l of GST–agarose was washed twice with PBS, added to the bacterial lysate and incubated for 4.5 h at 4°C with end-over-end rotation. The agarose was collected twice by centrifugation (first: 3 min, 2,500*g*, 4°C, second: 1 min 5,500*g*, 4°C) and washed five times with PBS. Proteins were eluted four times by incubation with GST–elution buffer (10 mM Glutathione, 50 mM Tris pH8) for 5 min each on ice and collection of the supernatant by centrifugation (30 min, 17,000*g*, 4°C). Protein concentration of all four individual fractions was determined.

H1299 cells were transfected with TRIM25. 24 h after transfection, cells were harvested and lysed in NP40 buffer (20 mM TRIS pH8, 400 mM NaCl, 1 mM EDTA, 1% Igepal CA-630, 0.1% SDS, 1% Protease Inhibitor Cocktail) for 10 min on ice. The lysate was cleared by centrifugation for 15 min at 17,000*g* at 4°C. Lysate corresponding to 450 *μ*g of proteins was diluted to 150 mM NaCl with No-salt buffer (20 mM Tris pH8, 1mM EDTA, 1% Igepal CA-630, 0.1%SDS, 1% Protease Inhibitor Cocktail). 25 mg freshly prepared GST-fused proteins (or GST for control) and 50 *μ*l of a 1:1 slurry of GST-agarose washed with PBS were added and incubated overnight at 4°C with end-over-end rotation. The agarose was collected by centrifugation and washed four times with wash buffer (20 mM Tris pH 8, 150 mM NaCl, 1 mM EDTA, 1% Igepal CA-630, 0.1% SDS). Proteins were eluted twice with 20 *μ*l 2x SDS sample buffer and heating for 10 min to 95°C each. The eluates were collected by centrifugation, combined, and separated by SDS–PAGE.

### Proximity ligation assay

$1 \times 10^4$ cells were plated on Chambered coverslips (Lab-Tek, Chambered 1.0 Borosilicate Coverglass System, Thermo Fisher Scientific). 24 h after plating the cells were treated with epoxomicin (7.5 *μ*M) or DMSO as a vehicle control, where indicated. After further 16 h (40 h after plating), cells were washed with PBS and fixed in ice-cold methanol/acetone solution (50/50 vol/vol) for 10 min. The methanol/acetone was removed and cells were washed three times with PBS. The following steps were carried out using Duolink in Situ Red Starter Kit Mouse/Rabbit (#DUO92101-1KT; Sigma-Aldrich) according to the manufacturer's protocol. Signals were acquired by laser-scanning confocal microscopy (confocal microscope platform Zeiss LSM900; Zeiss). Signals (red dots) were quantified per cell using the software ImageJ.

### RNA sequencing

Cells were lysed and RNA was prepared according to Yan et al (2015). Total RNA was sent to Novogene for sequencing. Fastq files were processed with CASAVA base calling and mapped against the human reference genome GRCH38/hg38 using HITSAT2 (Kim et al, 2019). Read numbers for each gene were quantified using featureCounts (Liao et al, 2014) and the reference gene annotation from Ensembl. All of this first analysis was done by Novogene. Normalization and differential expression analysis were performed using the R package DESeq2 (Love et al, 2014). For further analysis, CHIP-seq data (Zwart et al, 2011) (array express accession number E-MTAB-785) were used to select for p300 binding sites. Bam files for vehicle- and estradiol-treated MCF7 cells (two replicates each) were analyzed with R package ChIPpeakAnno. The consensus peaks

between vehicle and estradiol-treated cells and the peaks of the estradiol-treated cells were selected and the corresponding genes in a range of 20 kb to the transcription start site were used to filter the RNA-seq data for genes with a p300 binding site. RNA-seq data can be accessed via GEO (GSE215084).

### Statistical analysis

Mean values and standard deviations were calculated with Microsoft Excel. RNA-seq experiments, qRT-PCR assays, luciferase reporter assays, PLA and determinations of p300 degradation were performed with at least three biological replicates (exact numbers are in the figure legends). All other experiments were performed with at least two biological replicates.

## Data Availability

The RNA-seq data from this publication have been deposited to the GEO database (https://www.ncbi.nlm.nih.gov/geo) and assigned the accession number GSE215084.

## Supplementary Information

## Acknowledgements

We thank Christina Bauer for technical help; Andrew Cato, Satoshi Inoue, Ivan Dikic, Olivier Coux, Manuel Rodriguez, and Feng Zhang for plasmids, tools, and/or advice and the COST action ProteoCure CA20113; and the DFG grant No BL 526/12-1 for support. This work has furthermore received funding from the European Union's Horizon 2020 research and innovation programme under the Marie Sklodowska–Curie Grant Agreement 813599 (TRIM-NET). We acknowledge support by the KIT-Publication Fund of the Karlsruhe Institute of Technology.

### Author Contributions

S Elabd: data curation, formal analysis, investigation, and writing—review and editing.
E Pauletto: data curation, formal analysis, visualization, and writing—original draft, review, and editing.
V Solozobova: data curation, formal analysis, investigation, visualization, and writing—review and editing.
N Eickhoff: resources, formal analysis, and writing—review and editing.
N Padrao: resources, visualization, and writing—review and editing.
W Zwart: resources, formal analysis, and writing—review and editing.
C Blattner: conceptualization, resources, data curation, formal analysis, supervision, funding acquisition, validation, investigation, visualization, project administration, and writing—original draft, review, and editing.

## Conflict of Interest Statement

The authors declare that they have no conflict of interest.

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
