## [Reviewer comments · Life Science Alliance]

TRIM25 targets p300 for degradation

Seham Elabd, Eleonora Pauletto, Valeriya Solozobova, Nils Eickhoff, Nuno Padrao, Wilbert Zwart, and Christine Blattner

DOI: <https://doi.org/10.26508/lsa.202301980>

Corresponding author(s): Christine Blattner, Karlsruhe Institute of Technology and Wilbert Zwart, NKI

Review Timeline:

Submission Date:	2023-02-08
Editorial Decision:	2023-04-26
Revision Received:	2023-07-25
Editorial Decision:	2023-08-14
Revision Received:	2023-09-13
Editorial Decision:	2023-09-14
Revision Received:	2023-09-18
Accepted:	2023-09-19

Scientific Editor: Novella Guidi

Transaction Report:

April 26, 2023

Re: Life Science Alliance manuscript #LSA-2023-01980-T

Dr. Christine Blattner
Karlsruher Institute of Technology
Institute of Toxicology & Genetics
PO-Box 3640
Postfach 3640
Karlsruhe D-76021
Germany

Dear Dr. Blattner,

Thank you for submitting your manuscript entitled "TRIM25 targets p300 for degradation" to Life Science Alliance. The manuscript was assessed by expert reviewers, whose comments are appended to this letter. We invite you to submit a revised manuscript addressing the Reviewer comments.

Thank you for this interesting contribution to Life Science Alliance. We are looking forward to receiving your revised manuscript.

Sincerely,

B. MANUSCRIPT ORGANIZATION AND FORMATTING:

Reviewer #1 (Comments to the Authors (Required)):

Elabd S. et al. demonstrated an interesting and novel molecular mechanism that governs the turnover of the transcriptional cofactor p300 by TRIM25. Although the TRIM25-p300 interaction is required for p300 degradation, TRIM25 E3 activity is not utilized. Furthermore, autophagy could not explain TRIM25 targeting p300 degradation. This led the authors to their breakthrough finding, where they show that p300 is degraded in a dynein / microtubule-dependent manner. The manuscript is well written. All figures are clear and well organized. Here are several concerns/questions/suggestions that are expected to improve the overall of manuscript.

(1) Figure 1A, does the author see the same effect of TRIM25 overexpression on endogenous p300 as the exogenous one?

(2) Authors should be consistent throughout a manuscript using 'Figure' or 'Fig'.

(3) Page 18, line 24 (when TRIM25 was removed (Figure S4D).) I believe that the authors mistakenly refer to incorrect figures. The correct one is Figure 6C.

Cross-commenting section:

Dear Editor, (1) I agree with reviewer-2's comments that it would be preferable to use the same cell line(s) for Figure 1. It was reported that p300 promotes polyubiquitination of p53 (Science, 300(5617):342-4, 2003). The authors used H1299 cells (which did not express the p53 protein) to exclude the role of p53 and p300. (2) For Figure 2B, the authors demonstrate that TRIM25 does not affect ubi-p300 directly, as shown in Figure 2A. However, TRIM25 promotes p300 degradation. Also, I agree with reviewer-2 that the authors should label p300 with an arrow for Figure 2B. Although reviewer 2 suggested several experiments, such as proximity ligation assay to further prove the physical interaction between Dynein and p300, it will improve the quality of this manuscript. The authors have addressed that TRIM25 links p300 to dynein allowing a dynein/microtubule-dependent degradation in this manuscript. Finally, it should allow the authors to respond to the questions raised by reviewer-2.

Reviewer #2 (Comments to the Authors (Required)):

In this manuscript, the authors perform a set of experiments to demonstrate that TRIM25, a member of the TRIM family and a ubiquitin E3 ligase, antagonizes the stability of the transcriptional co-regulator p300 in a manner that is uncoupled from its ubiquitin E3 ligase activity. They provide evidence that TRIM25, which interacts with both p300 and Dynein also promotes p300's association with Dynein, and suggest that the latter directs p300 to the proteasomes. Finally, the authors demonstrate that TRIM25 silencing has an impact on the expression of p300 target genes.

Minor comments:

Figure 1:

Here, using multiple cell systems, the authors demonstrate that TRIM25 expression has an antagonistic effect on p300 protein levels. To this end, they used H1299 cells to overexpress TRIM25 or HCT116 cells to silence TRIM25 expression by siRNA. Why did the authors use one type of cell line for overexpression and another cell line for siRNA silencing? It would be preferable to use the same cell line for both analyses.

They also used TRIM25 knockout MEFs, as well as MCF7 cells where TRIM25 expression was silenced either by CRISPR or shRNA.

All these experiments clearly demonstrate that TRIM25 silencing has a positive impact on p300 protein levels (the authors also show in TRIM25 -/- MEFs that p300 mRNA transcript levels are not affected).

Figure 2:

Here, the authors test whether TRIM25 promotes p300 ubiquitylation. In 2A, they perform a His-pulldown under denaturing conditions which convincingly demonstrate that TRIM25 does not induce p300 ubiquitylation. In 2B, the authors perform an immunoprecipitation assay where they pull down p300 to detect its ubiquitylated forms by Western blot. Unfortunately, this figure is confusing and not informative. Which band represents p300? Why is there a band in the lane where a non-specific IgG was used? Did the authors treat these cells with a proteasome inhibitor to stabilize the ubiquitylated p300 forms? The gel is very dirty and the ubiquitylated forms are not easy to see. A better and more representative experiments needs to be shown here.

Figure 3:

To gain more insight into the mechanisms of TRIM25-mediated p300 destabilization, the authors have made several deletion (truncation) mutations on TRIM25. Using these mutants, they conclude that only the "B-box and coiled coil" domains of TRIM25 were able to antagonize p300 protein stability and that the RING domain was dispensable. Fig 3A indeed shows that the Delta2 mutant (B-box coiled coil) promotes almost full degradation of p300. However, both Delta 1 and Delta 3 mutants are also able to initiate p300 degradation to a certain extent, though not as efficient as Delta 2. What is the authors' explanation for this observation? It seems like the RING domain also has some activity here. This Western blot is also lacking the loading controls.

Fig 3B clearly demonstrates that a catalytically-dead TRIM25 mutant was still capable of promoting p300 degradation.

Here, in order to demonstrate that p300 is indeed degraded in a proteasome-dependent manner when TRIM25 or its catalytically-dead mutant is overexpressed, the authors should use a proteasome inhibitor and also a proteasome activator (i.e. PD169316) and perform a cycloheximide chase experiment to follow p300 half life. This would greatly help establish that the proteasome is indeed responsible for TRIM25-mediated destabilization of p300.

Figure 4:

In this figure, the authors further explore the mechanisms of TRIM25-mediated p300 degradation. I have a few comments about this figure.

4A: Here, the authors perform TRIM25 overexpression. However, in the main text (lines 138, 139), they sound as if they are performing TRIM25 silencing or downregulation. This should be corrected.

Line 142 of main text: The authors should avoid stating that "ubiquitin-proteasome action could not explain TRIM25's capacity to target p300 for degradation" because clearly the message the authors want to convey is that p300 is targeted to proteasomes via a cooperation between TRIM25 and Dynein. Therefore, the proteasome action is indeed involved and needed. Plus, data in Fig 2A suggest that the ubiquitylated forms of p300 may be more prone to destabilization by TRIM25, although TRIM25 itself does not promote p300's ubiquitylation.

4H: This co-IP experiment is very difficult to interpret and does not look very convincing. It should be replaced with a better image / experiment. Which band on the left (IP panel) is p300? Why don't we see any p300 in the 1st lane? Because it is fully degraded? In that case why do we see Dynein coming down with it (lane 1)?

Importantly, in order to further prove Dynein & p300 physical interaction, a complementary approach such as proximity ligation assay would be nice to include. This would also allow visualization of the interaction signals in situ and help the authors determine where these interactions actually take place within the cell (i.e. perhaps towards the pericentriolar region?).

Figures 6 & 7:

Here, the authors show the impact of TRIM25 on p300-driven gene expression. To this end, they either transfect HEK293T cells with an MMTV reporter, along with AR or GR (and treat cells with either DHT or Dexa) or use TRIM25^{-/-} MEFs. These results indicate that hormone-induced and p300-driven gene transcription is indeed antagonized by TRIM25.

RNA sequencing in ER-dependent MCF7 cells in which TRIM25 is knocked out, integrated with a prior p300 ChIP-Seq analysis in estrogen treated MCF7 cells allowed the authors to focus on 4402 direct p300 targets, 148 of which were induced upon TRIM25 silencing.

Figure 8:

Once again, in order to support their model, the authors may wish to provide additional evidence for physical interactions between various partners, i.e. "p300" and "the proteasome" by performing proximity ligation assays. If they performed this experiment, would they see p300/proteasome interaction signals localized to the pericentriolar region, as they propose in their model? What if they performed the same PLA analysis upon TRIM25 overexpression or silencing, or in the presence of a

proteasome inhibitor? In each condition, they could quantify the interaction signals and also determine their subcellular localization.

Reviewer #1:

Elabd S. et al. demonstrated an interesting and novel molecular mechanism that governs the turnover of the transcriptional cofactor p300 by TRIM25. Although the TRIM25-p300 interaction is required for p300 degradation, TRIM25 E3 activity is not utilized. Furthermore, autophagy could not explain TRIM25 targeting p300 degradation. This led the authors to their breakthrough finding, where they show that p300 is degraded in a dynein / microtubule-dependent manner. The manuscript is well written. All figures are clear and well organized.

Answer: We thank the reviewer for the time invested in evaluating our work. We were delighted to see the reviewer finds our manuscript of interest, and we thank the reviewer for the constructive and helpful suggestions that clearly helped us to further improve the quality of our work. A response to all points raised, is listed below.

Here are several concerns/questions/suggestions that are expected to improve the overall of manuscript.

(1) Figure 1A, does the author see the same effect of TRIM25 overexpression on endogenous p300 as the exogenous one?

Answer: Yes, we do. In the new version, we have modified Figure 1. In this new figure 1, we now provide new data. In part A.I, we have overexpressed p300 and TRIM25. In part A.II, we have only overexpressed TRIM25 in H1299 and in part A.III, again with employing H1299 cells, we have downregulated TRIM25. By using all three variations of overexpression/downregulation, we consistently observed downregulation of p300 by TRIM25.

(2) Authors should be consistent throughout a manuscript using 'Figure' or 'Fig'.

Answer: We are very sorry for having overseen this. Yes, we should be consistent and we have now corrected these inconsistencies, replacing all "Fig"s to "Figure".

(3)

Page 18, line 24 (when TRIM25 was removed (Figure S4D).) I believe that the authors mistakenly refer to incorrect figures. The correct one is Figure 6C.

Answer: We thank the reviewer for bringing this issue to our attention, and we apologize for the misunderstanding. What we wanted to say was that the downregulation of TRIM25 for the cell lines that were used for this experiment is displayed in (former) Figure S4D (which is now Supplementary figure 6D). We now rephrased this sentence, to resolve the unclarity:

("RNA-seq results were successfully and independently validated by RT-qPCR using two independent

MCF7 TRIM25 knock-out lines (the reduction in TRIM25 is shown in Supplementary figure 6D)....")

Cross-commenting section:

(1) I agree with reviewer-2's comments that it would be preferable to use the same cell line(s) for Figure 1.

Answer: We thank the reviewers for this advice. Our intention was to show the universal observation of this regulation of p300 by TRIM25 and therefore, we displayed several different cell lines. Given the comments received from the reviewers on this issue, this design choice possibly confused the reader. In the revised version, we have removed the HCT116 cells (to reduce the number of different cell lines at least by one) and performed the downregulation of TRIM25 in the H1299 cell line. In addition, we now added data on the overexpression of TRIM25 in the H1299 cell line. This way, we have overexpression of p300 and TRIM25, overexpression of only TRIM25 and downregulation of TRIM25 all in one cell line (in H1299 cells). To highlight these biological features of TRIM25 are consistent over different model systems derived from different tissues, we still included the MEF and MCF7 cells in Figure 1, showing that the regulation of p300 by TRIM25 is a more general phenomenon.

It was reported that p300 promotes polyubiquitination of p53 (Science, 300(5617):342-4, 2003). The authors used H1299 cells (which did not express the p53 protein) to exclude the role of p53 and p300. (2) For Figure 2B, the authors demonstrate that TRIM25 does not affect ubi-p300 directly, as shown in Figure 2A. However, TRIM25 promotes p300 degradation.

Also, I agree with reviewer-2 that the authors should label p300 with an arrow for Figure 2B.

Answer: We thank the reviewer for the suggestion. We have added an arrow where the p300 signal is.

Although reviewer 2 suggested several experiments, such as proximity ligation assay to further prove the physical interaction between Dynein and p300, it will improve the quality of this manuscript.

Answer: We thank the reviewers for this suggestion. For revised manuscript, we have now performed PLA and have added the results to the manuscript (please see Figure 4 and Supplementary figure 5). These results further strengthened our original observations, through orthogonal methods, which clearly strengthened our manuscript further.

The authors have addressed that TRIM25 links p300 to dynein allowing a dynein/microtubule-dependent degradation in this manuscript.

Answer: We thank the reviewer for this assessment.

Finally, it should allow the authors to respond to the questions raised by reviewer-2.

Answer: please find below our answers to reviewer-2.

Reviewer #2:

In this manuscript, the authors perform a set of experiments to demonstrate that TRIM25, a member of the TRIM family and a ubiquitin E3 ligase, antagonizes the stability of the transcriptional co-regulator p300 in a manner that is uncoupled from its ubiquitin E3 ligase activity. They provide evidence that TRIM25, which interacts with both p300 and Dynein also promotes p300's association with Dynein, and suggest that the latter directs p300 to the proteasomes. Finally, the authors demonstrate that TRIM25 silencing has an impact on the expression of p300 target genes.

Answer: We thank the reviewer for the time spent in reviewing our manuscript, and for the highly constructive and helpful comments that clearly enabled us to further improve the quality of our manuscript. We did our utmost best to address all issues that were raised, as listed point-by-point below.

Minor comments:

Figure 1: Here, using multiple cell systems, the authors demonstrate that TRIM25 expression has an antagonistic effect on p300 protein levels. To this end, they used H1299 cells to overexpress TRIM25 or HCT116 cells to silence TRIM25 expression by siRNA. Why did the authors use one type of cell line for overexpression and another cell line for siRNA silencing? It would be preferable to use the same cell line for both analyses. They also used TRIM25 knockout MEFs, as well as MCF7 cells where TRIM25 expression was silenced either by CRIPSR or shRNA.

Answer: We thank the reviewer for this suggestion. We used several cell lines to show that this regulation of p300 by TRIM25 is not restricted to a specific cell line, but rather that this observation of TRIM25-mediated p300 regulation is a more general features shared by different models from different organs. Given the reviewer comments on this issue, we now realize this design choice may have made things unnecessarily complicated. Following the reviewer's advice, we made Figure 1 more consistent in the cell line usage, by removing the HCT116 cells and performing overexpression and downregulation of TRIM25 in H1299 cells so that we now have overexpression of p300 and TRIM25, overexpression of TRIM25 alone and downregulation of TRIM25 in one and the same cell line (Figure 1A). To highlight the general feature of TRIM25-mediated p300 degradation, and demonstrate these effects are also seen in other cell line models, we retained the results generated with MEFs and MCF-7 cells as part of Figure 1.

All these experiments clearly demonstrate that TRIM25 silencing has a positive impact on p300 protein levels (the authors also show in TRIM25 $-/-$ MEFs that p300 mRNA transcript levels are not affected).

Answer: Thank you. Yes, this is what we see.

Figure 2: Here, the authors test whether TRIM25 promotes p300 ubiquitylation. In 2A, they perform a His-pulldown under denaturing conditions which convincingly demonstrate that TRIM25 does not induce p300 ubiquitylation. In 2B, the authors perform an immunoprecipitation assay where they pull down p300 to detect its ubiquitylated forms by Western blot. Unfortunately, this figure is confusing and not informative. Which band represents p300? Why is there a band in the lane where a non-specific IgG was used? Did the authors treat these cells with a proteasome inhibitor to stabilize

the ubiquitylated p300 forms? The gel is very dirty and the ubiquitylated forms are not easy to see. A better and more representative experiments needs to be shown here.

Answer: We are pleased to see the reviewer finds Figure 2A convincing, and we thank the reviewer for highlighting these improvement points for Figure 2B. Following the reviewer's advice, we now replaced these images with better quality visualizations. Of note, we first hybridized the IP with an anti-ubiquitin antibody. The antibody works very well as we now show in the input. However, since we have done the IP for p300 and p300 is not much ubiquitinated the ubiquitinated forms are again not easily seen, unfortunately. Of note, these observations fully support our conclusion that TRIM25 does not ubiquitinate p300. After hybridization with an anti-ubiquitin-antibody, we hybridized the blot with the anti-HA antibody for HA-p300. However, since p300 is hard to detect and now it was hybridized for p300 after the hybridization for ubiquitin, it is again somewhat dirty and the p300 signal is again weak (as is expected, being a negative result that is important for the rest of the study). For better visualization what the p300 signal is, we have added an arrow where the p300 signal is.

Figure 3: To gain more insight into the mechanisms of TRIM25-mediated p300 destabilization, the authors have made several deletion (truncation) mutations on TRIM25. Using these mutants, they conclude that only the "B-box and coiled coil" domains of TRIM25 were able to antagonize p300 protein stability and that the RING domain was dispensable. Fig 3A indeed shows that the Delta2 mutant (B-box coiled coil) promotes almost full degradation of p300. However, both Delta 1 and Delta 3 mutants are also able to initiate p300 degradation to a certain extent, though not as efficient as Delta 2. What is the authors' explanation for this observation? It seems like the RING domain also has some activity here. This Western blot is also lacking the loading controls.

Answer: We thank the reviewers for putting our attention to this figure. Following the reviewers recommendation, we wanted to replace these data with a new figure, now providing a proper loading control. However, when we repeated this experiment, we realized that the different TRIM25 versions were expressed at very different levels, impacting the conclusions drawn from these analyses. Particularly the RING domain was expressed at very low levels. We therefore attempted to adjust the expression of all the TRIM25 versions by transfecting more of the RING- and also of the C-terminal fragment. Yet, this required to reduce the ratio of TRIM25:p300 as we could not transfect less p300 as we were (always) already close to the detection limit of p300. However, reducing the ratio of TRIM25:p300 had the consequence, that we could no longer see the targeting of p300 for degradation by TRIM25. So we had to decide to either accept unequal expression levels of TRIM25 constructs (with the consequence that we may oversee the action of some of the mutants due to their low expression levels) or loose the degradation of p300 because the ratio of TRIM25:p300 comes below the threshold for action. Of note, these analyses did confirm our previous observations that the C-Box/coiled-coiled region, a TRIM25 construct that was well expressed, was solely sufficient for targeting p300 for degradation. Nevertheless, as the above-mentioned technical limitations prevented us to draw solid conclusions for all the constructs, we decided to remove this figure from the manuscript (and add the part B of this figure to the previous figure (Figure 2)).

Fig 3B clearly demonstrates that a catalytically-dead TRIM25 mutant was still capable of promoting p300 degradation. Here, in order to demonstrate that p300 is indeed degraded in a proteasome-dependent manner when TRIM25 or its catalytically-dead mutant is overexpressed, the authors should use a proteasome inhibitor and also a proteasome activator (i.e. PD169316) and perform a

cycloheximide chase experiment to follow p300 half life. This would greatly help establish that the proteasome is indeed responsible for TRIM25-mediated destabilization of p300.

Answer: We thank the reviewers very much for this suggestion. We performed half-life experiments with MCF7 cells in the presence and absence of epoxomicin and PD169316 and also with H1299 cells that had been transfected with p300 alone or with p300 and TRIM25. Unfortunately, PD169316 did not give conclusive results, neither in MCF7 cells nor in H1299 cells. We therefore did not include these experiments in the revised version of the manuscript. However, by treating the cells with epoxomicin, we did robustly and consistently observe the dependence of TRIM25-mediated p300 degradation on cellular proteasomes. The rationale for us, to consider these data suitable to support this conclusion, is the following: When we transfected only p300 into the cells, the p300 protein was completely stable and we could not see any further increase in p300 levels by epoxomicin. In contrast, when we transfected p300 together with TRIM25, p300 protein was degraded, showing the TRIM25-dependence of p300 degradation. When we then blocked proteasomes by treating the cells with epoxomicin, the p300 protein was completely stable, even in the presence of TRIM25. These results are shown in Supplementary figure 3 of the revised version of the manuscript.

Figure 4: In this figure, the authors further explore the mechanisms of TRIM25-mediated p300 degradation. I have a few comments about this figure.

4A: Here, the authors perform TRIM25 overexpression. However, in the main text (lines 138, 139), they sound as if they are performing TRIM25 silencing or downregulation. This should be corrected.

Answer: The reviewer is completely right, and we apologize for this mistake. We have corrected the text accordingly.

Line 142 of main text: The authors should avoid stating that "ubiquitin-proteasome action could not explain TRIM25's capacity to target p300 for degradation" because clearly the message the authors want to convey is that p300 is targeted to proteasomes via a cooperation between TRIM25 and Dynein. Therefore, the proteasome action is indeed involved and needed. Plus, data in Fig 2A suggest that the ubiquitylated forms of p300 may be more prone to destabilization by TRIM25, although TRIM25 itself does not promote p300's ubiquitylation.

Answer: Yes, the reviewer is right and our previous statement was misleading. We have therefore changed this sentence into: "Neither autophagy nor TRIM25-mediated ubiquitination of p300 could explain TRIM25's capacity to target p300 for degradation."

4H: This co-IP experiment is very difficult to interpret and does not look very convincing. It should be replaced with a better image / experiment. Which band on the left (IP panel) is p300? Why don't we see any p300 in the 1st lane? Because it is fully degraded? In that case why do we see Dynein coming down with it (lane 1)?

Answer: We apologize for the unclarity, and thank the reviewer for bringing this issue to our attention. It is extremely difficult to perform co-immunoprecipitations with p300. We have tried with two different p300-specific antibodies (most available antibodies also recognize CBP and because of the higher levels of CBP (which is not regulated by TRIM25), it masks the regulation of p300. These antibodies could therefore not be used in our study. In addition to the technical limitations of commercially available antibodies, the level of p300 in cells is extremely low and difficult to detect

both in IP and Western blots. We are very grateful for the suggestion of the reviewers to perform PLAs, which are obviously much more sensitive and which allowed us to support our co-immunoprecipitation of p300 and dynein by orthogonal methods. We hope that in the connection with the support by the PLA, the reviewers can accept the co-immunoprecipitation of p300 and dynein that is contained in the manuscript.

Importantly, in order to further prove Dynein & p300 physical interaction, a complementary approach such as proximity ligation assay would be nice to include. This would also allow visualization of the interaction signals in situ and help the authors determine where these interactions actually take place within the cell (i.e. perhaps towards the pericentriolar region?).

Answer: We thank the reviewer for this excellent suggestion, that clearly helped us to further improve the quality of our work. We have performed PLA and included the data in the new version of the manuscript (Figure 4 and Supplementary figure 5)

Figures 6 & 7: Here, the authors show the impact of TRIM25 on p300-driven gene expression. To this end, they either transfect HEK293T cells with an MMTV reporter, along with AR or GR (and treat cells with either DHT or Dexa) or use TRIM25^{-/-} MEFs. These results indicate that hormone-induced and p300-driven gene transcription is indeed antagonized by TRIM25. RNA sequencing in ER-dependent MCF7 cells in which TRIM25 is knocked out, integrated with a prior p300 ChIP-Seq analysis in estrogen treated MCF7 cells allowed the authors to focus on 4402 direct p300 targets, 148 of which were induced upon TRIM25 silencing.

Figure 8: Once again, in order to support their model, the authors may wish to provide additional evidence for physical interactions between various partners, i.e. "p300 "and "the proteasome" by performing proximity ligation assays. If they performed this experiment, would they see p300/proteasome interaction signals localized to the pericentriolar region, as they propose in their model? What if they performed the same PLA analysis upon TRIM25 overexpression or silencing, or in the presence of a proteasome inhibitor? In each condition, they could quantify the interaction signals and also determine their subcellular localization.

Answer: We again thank the reviewer for this excellent suggestion. We have performed PLA of non-targeted and knock-out cells for p300 and dynein and for p300 and the proteasome with and without epoxomicin. In the presence of epoxomicin, we indeed see localization at the perinuclear space. These experiments are shown Supplementary figure 5.

August 14, 2023

Re: Life Science Alliance manuscript #LSA-2023-01980-TR

Dr. Christine Blattner
Karlsruher Institute of Technology
Institute for Biological and Chemical Systems - Biological Information Processing
PO-Box 3640
Postfach 3640
Karlsruhe D-76021
Germany

Dear Dr. Blattner,

Thank you for submitting your revised manuscript entitled "TRIM25 targets p300 for degradation" to Life Science Alliance. The manuscript has been seen by the original reviewers whose comments are appended below. While the reviewers continue to be overall positive about the work in terms of its suitability for Life Science Alliance, some important issues remain.

Our general policy is that papers are considered through only one revision cycle; however, given that the suggested changes are relatively minor, we are open to one additional short round of revision. Please note that I will expect to make a final decision without additional reviewer input upon re-submission.

Please submit the final revision within one month, along with a letter that includes a point by point response to the remaining reviewer comments.

To upload the revised version of your manuscript, please log in to your account: <https://lsa.msubmit.net/cgi-bin/main.plex>
You will be guided to complete the submission of your revised manuscript and to fill in all necessary information.

B. MANUSCRIPT ORGANIZATION AND FORMATTING:

Sincerely,

Reviewer #1 (Comments to the Authors (Required)):

The current version of Elabd et al is much improved. The authors have largely addressed the comments/questions raised with

new data and the discussion, which made this study stronger and better suited for publication.

Reviewer #2 (Comments to the Authors (Required)):

The authors have now address most of my concerns and comments.

However, I am still not convinced by the pulldown experiment in Fig 2B. I am really not sure if the so called bands on top the gel in the ubiquitin blot really represent the ubiquitylated forms of p300. Some important controls are still lacking here, i.e. +/- proteasome inhibitor conditions, or a lane where His-ubiquitin overexpression was not performed. Presentation of the uncropped gel could also help to a certain extent. In light of the rest of the data that the authors present (i.e. Fig 2A), I think that this figure should be omitted from the final version of the manuscript, unless the authors cannot replace it with a better one that includes the proper controls or with a PLA experiment to probe p300-ubiquitin conjugation.

Also, as a response to my comment to Fig 3B, the authors state that "Unfortunately, PD169316 did not give conclusive results, neither in MCF7 cells nor in H1299 cells. We therefore did not include these experiments in the revised version of the manuscript". This is somewhat of a concern. I wonder whether in these experiments the authors used a positive control (another substrate whose turnover is dependent on the proteasome) to see if the drug indeed works well and the treatments were properly performed.

In all PLA experiments, the authors must include proper negative controls where only one of the antibody pairs is used. For this purpose, for example in Fig 4B, they include a negative control p300-IgG. However, the other negative control Dynein-IgG is missing. This is important and the same goes for Fig 4F.

Some of the supplementary figures are mislabelled, for example Supplementary fFigure 4 is wrongly labelled as Supplementary figure 3, etc.

Rebuttal letter (addressing the reviewers' comments point by point)
Life Science Alliance manuscript #LSA-2023-01980-TR

Reviewer #1

The current version of Elabd et al is much improved. The authors have largely addressed the comments/questions raised with new data and the discussion, which made this study stronger and better suited for publication.

Answer: We thank the reviewer for the time invested in evaluating our work and the revised manuscript. We were delighted to see the reviewer finds our revised manuscript improved and better suited for publication.

Reviewer #2

The authors have now address most of my concerns and comments.

Answer: We thank the reviewer for the time spent in reviewing our original and revised manuscript, and for the highly constructive and helpful comments that clearly enabled us to further improve the quality of our manuscript. We did our utmost best to address all issues that were raised, as listed point-by-point below.

However, I am still not convinced by the pulldown experiment in Fig 2B. I am really not sure if the so called bands on top the gel in the ubiquitin blot really represent the ubiquitylated forms of p300. Some important controls are still lacking here, i.e. +/- proteasome inhibitor conditions, or a lane where His-ubiquitin overexpression was not performed. Presentation of the uncropped gel could also help to a certain extent. In light of the rest of the data that the authors present (i.e. Fig 2A), I think that this figure should be omitted from the final version of the manuscript, unless the authors cannot replace it with a better one that includes the proper controls or with a PLA experiment to probe p300-ubiquitin conjugation.

Answer: We thank the reviewer for this suggestion. Following the reviewer's advice, we have now replaced Fig. 2B with a PLA experiment. As we show in (the new) supplementary figure 2, there is a basic ubiquitination of p300. However, this basic ubiquitination does not change when TRIM25 is knocked-out or when we treat the cells with the proteasome inhibitor epoxomicin. For comparison, we have also done a PLA for ubiquitin and p53, a protein that is well-known as a target for polyubiquitination and proteasomal degradation. As the reviewer will see, ubiquitination of p53 was magnitudes stronger than that of p300. We hypothesize that the low basic ubiquitination of p300 serves for other regulatory issues but is not linked to proteasomal degradation.

Also, as a response to my comment to Fig 3B, the authors state that "Unfortunately, PD169316 did not give conclusive results, neither in MCF7 cells nor in H1299 cells. We therefore did not include these experiments in the revised version of the manuscript". This is somewhat of a concern. I wonder whether in these experiments the authors used a positive control (another substrate whose turnover is dependent on the proteasome) to see if the drug indeed works well and the treatments were properly performed.

Answer: We apologize for the unclarity in the description of our results. We have now put a lot of additional emphasis into this point of concern. The reviewer's concerns motivated us to re-evaluate all our old experiments. If at all, a small amount of stabilization could be detected, but this was not

consistent for all experiments. As we expected that PD169316 should enhance proteolysis, yet no positive controls were included in our first experiments, we now repeated the experiments with MCF-7 cells in biological triplicates and this time we also analyzed the degradation of the tumor suppressor protein p53, a bona-fide candidate for proteasomal degradation. We also compared our conditions to the ones from Leestemaker et al., 2017 and made sure that we have the same cells (MCF-7), the same incubation time (16 hours) and the same dose range (5-10 μ M). But again, we did not see increased proteolysis, neither for p300 nor for p53 in the presence of PD169316. To rule out that the compound had lost its efficacy (although we had bought it new), we purchased another independent vial of the PD169316, in solubilized state. Using this compound, we performed another three independent biological replicates within 2 weeks to ensure that the compound would not be degraded before completion of our experiments. Nevertheless, we again had the same result, which is no change in p53 half-life (now serving as control) and if at all, a small stabilization of p300 in some experiments. We also searched the literature for other publications where this compound was used for increased proteolysis, but did not really find a clear outcome. We therefore conclude that this compound might only stimulate proteolysis under very specific conditions that we were not able to elucidate in the short time frame of the revision.

In all PLA experiments, the authors must include proper negative controls where only one of the antibody pairs is used. For this purpose, for example in Fig 4B, they include a negative control p300-IgG. However, the other negative control Dynein-IgG is missing. This is important and the same goes for Fig 4F.

Answer: *We apologize for this omission, which has now been included in the revised version of our manuscript. We have now provided further controls where only antibodies targeted against p300, dynein or alpha-7 and also against p53 and ubiquitin were used together with IgG from the corresponding species.*

Some of the supplementary figures are mislabelled, for example Supplementary Figure 4 is wrongly labelled as Supplementary figure 3, etc.

Answer: *The reviewer is correct. During the revision process, apparently some derangement occurred that slipped through prior to submission. We deeply apologize for this confusion. We now ensured that all figures are labelled correctly and introduce the figures and references to the figures now into the right order.*

September 14, 2023

RE: Life Science Alliance Manuscript #LSA-2023-01980-TRR

Dr. Christine Blattner
Karlsruher Institute of Technology
Institute for Biological and Chemical Systems - Biological Information Processing
PO-Box 3640
Postfach 3640
Karlsruhe D-76021
Germany

Dear Dr. Blattner,

Thank you for submitting your revised manuscript entitled "TRIM25 targets p300 for degradation". We would be happy to publish your paper in Life Science Alliance pending final revisions necessary to meet our formatting guidelines.

- please add a callout for Fig S6B to your main manuscript text
- please add the Twitter handle of your host institute/organization as well as your own or/and one of the authors in our system
- the supplementary material and methods should be incorporated into the main Materials and Methods section. Same for the Supplemental Reference; should be in the main Reference list
- you may want to consider uploading Figure 7 as a Graphical Abstract, but this is up to you

A. FINAL FILES:

B. MANUSCRIPT ORGANIZATION AND FORMATTING:

**Submission of a paper that does not conform to Life Science Alliance guidelines will delay the acceptance of your

manuscript.**

The license to publish form must be signed before your manuscript can be sent to production. A link to the electronic license to publish form will be sent to the corresponding author only. Please take a moment to check your funder requirements.

Sincerely,

September 19, 2023

RE: Life Science Alliance Manuscript #LSA-2023-01980-TRRR

Dr. Christine Blattner
Karlsruher Institute of Technology
Institute for Biological and Chemical Systems - Biological Information Processing
PO-Box 3640
Postfach 3640
Karlsruhe D-76021
Germany

Dear Dr. Blattner,

Thank you for submitting your Research Article entitled "TRIM25 targets p300 for degradation". It is a pleasure to let you know that your manuscript is now accepted for publication in Life Science Alliance. Congratulations on this interesting work.

DISTRIBUTION OF MATERIALS:

Again, congratulations on a very nice paper. I hope you found the review process to be constructive and are pleased with how the manuscript was handled editorially. We look forward to future exciting submissions from your lab.

Sincerely,
